# Constant Rate Schedule: Constant-Rate Distributional Change for Efficient Training and Sampling in Diffusion Models

## Abstract

We propose a noise schedule that ensures a constant rate of change in the probability distribution of diffused data throughout the diffusion process. To obtain this noise schedule, we measure the rate of change in the probability distribution of the forward process and use it to determine the noise schedule before training diffusion models. The functional form of the noise schedule is automatically determined and tailored to each dataset and type of diffusion model. We evaluate the effectiveness of our noise schedule on unconditional and class-conditional image generation tasks using the LSUN (bedroom/church/cat/horse), ImageNet, and FFHQ datasets. Through extensive experiments, we confirmed that our noise schedule broadly improves the performance of the diffusion models regardless of the dataset, sampler, number of function evaluations, or type of diffusion model.

## 1 Introduction

Image generation is one of the most challenging tasks in computer vision, and a variety of deep generative models have been proposed. Generative adversarial networks (GANs) (Goodfellow et al., 2014) have long been the leading models for high-quality image generation. These generative models achieved success across a wide range of fields beyond image generation, such as audio (van den Oord et al., 2016; Kong et al., 2021) and 3D-point cloud generation (Yang et al., 2019).

The performance of generative models is measured using three metrics: sampling speed, sample quality, and mode coverage (Xiao et al., 2022). Despite the extensive research conducted, satisfying these requirements simultaneously is challenging. For example, GANs have strengths in terms of sampling speed and sample quality, but exhibit a weakness in mode coverage.

Denoising diffusion probabilistic models (DDPMs) (Sohl-Dickstein et al., 2015; Ho et al., 2020) are new generative models, that achieve high sample quality and mode coverage. DDPMs are latent variable models that use a Markov chain to gradually add Gaussian noise to the data, called the forward process. At the end of the forward process, the data are completely destroyed to pure noise. The reverse process is introduced to restore the data from the pure noise by tracing the forward process in the reverse direction. The reverse process uses a Markov chain as well, and a deep neural network is trained to sequentially remove the added noise in latent variables. Although the annealing nature of DDPMs contributes to the high mode coverage, the iterative denoising steps result in slow sampling speed. The number of steps in the reverse process is set to as much as one thousand, as described in Ho et al. (2020).

In this work, to reduce the number of steps in the reverse process, we propose a noise schedule that ensures a constant rate of change in the probability distribution of diffused data throughout the diffusion process, which we refer to as CRS. As previously stated, the reverse process needs to accurately trace the forward process to restore the data from pure noise. This motivates us to search for noise schedules to enhance the traceability of the forward process. Figure 1 depicts a toy example of diffused data distributions with three data points in one-dimensional data space (see Appendix A for details). In this example, the diffusion process can be divided into three regions. The first is an area where the probability distributions hardly change, and the noise level can be changed rapidly ($\alpha \lesssim 0.6$). The second is an area where three modes corresponding to the data points emerge, and careful adjustment of the noise level is necessary to enhance mode coverage ($0.6 \lesssim \alpha \lesssim 0.97$).

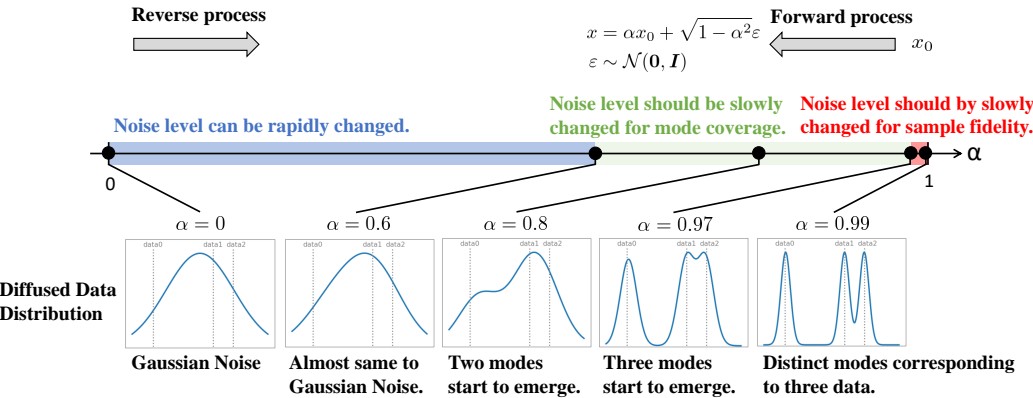

Figure 1: A toy example of diffused data distributions with three data points in one dimensional data space. The probability distributions hardly change when $\alpha \lesssim 0.6$, and we can rapidly change the noise level. Three modes corresponding to the data points emerge when $0.6 \lesssim \alpha \lesssim 0.97$, and noise level should be changed slowly for mode coverage. The three modes become distinct when $\alpha \gtrsim 0.97$, requiring careful control of the noise level for sample fidelity.

The third is an area where the peaks of each mode become distinct, and the noise level should be changed slowly for sample fidelity ($\alpha \gtrsim 0.97$). As evident from this example, the diffusion process exhibits regions with rapid and slow changes in the probability distributions. To obtain CRS, we measure the rate of change in the probability distribution of the forward process, which we assume represents the traceability. We then determine the noise schedule so that the rate of change remains constant throughout the forward process before training diffusion models. The functional form of CRS is automatically determined for ensuring the constancy of probability-distributional change. Therefore, it is not necessary to pre-define the noise-schedule function, such as linear (Ho et al., 2020) and cosine (Nichol & Dhariwal, 2021). CRS is considered a generalized version of Song & Ermon (2020), which determines the noise schedule so that a constant overlap between consecutive probability distributions in the forward process is achieved. Through extensive experiments, we conclude that CRS broadly improves the performance of image generation in diffusion models.

We make the following contributions:

1. We propose an efficient noise schedule for training and sampling, which we refer to as CRS, to boost the performance of diffusion models. CRS enables us to automatically derive a noise schedule tailored to the target dataset and type of diffusion model.

2. CRS holds significant value as a general framework for customizing noise schedules. Once an appropriate distance metric for measuring the probability-distributional change is defined, the corresponding noise schedule is systematically obtained.

3. We find that the efficient noise schedule depends on both the target dataset and the type of diffusion model. In particular, the efficient noise schedule in latent-space diffusion models is very different from conventionally used ones.

4. Through extensive experiments, we confirm that CRS broadly improves the performance of image generation regardless of the dataset, sampler, number of function evaluations (NFE), or type of diffusion model.

The rest of this paper is organized as follows: Section 2 explains related work on reducing the number of steps in image generation. Section 3 briefly introduces diffusion models. Section 4 presents CRS. Section 5 presents the experimental results to confirm the superiority of CRS. Section 6 presents a discussion, and Section 7 concludes this paper.

## 2 RELATED WORK

Various methods have been proposed to reduce the number of steps in the reverse process. Most methods can be categorized into four strategies: conditional generation, data dimensionality reduction, samplers, and noise schedules.

**Conditional generation:** This strategy reduces the number of steps by eliminating unnecessary sample diversity. The attributes of the samples to be generated are specified through conditioning, such as class and text (Dhariwal & Nichol, 2021; Ho & Salimans, 2021; Rombach et al., 2022; Saharia et al., 2022; Nichol et al., 2022; Podell et al., 2023).

**Dimensionality reduction of data:** The latent diffusion model (LDM) (Rombach et al., 2022; Podell et al., 2023) utilizes a pre-trained autoencoder to reduce the dimensionality of the data, and diffusion models are applied to the reduced data space. This strategy not only reduces the number of necessary steps but also decreases the processing time per step.

**Samplers:** This strategy devises update rules for latent variables in the reverse process. The Denoising Diffusion Implicit Model (DDIM) (Song et al., 2021a) introduces a non-Markov chain to generalize DDPMs and proposes a deterministic sampler. From the perspective of score-based matching, Song & Ermon (2019; 2020); Song et al. (2021b) developed continuous-time diffusion models. In one of these studies (Song et al., 2021b), the time evolution of latent variables was formulated using a stochastic differential equation (SDE) and an ordinary differential equation (ODE). They also proposed using general solvers of the SDE and ODE for sampling in diffusion models. Subsequent studies have actively investigated customizing these general solvers for diffusion models (Jolicoeur-Martineau et al., 2021; Liu et al., 2022; Lu et al., 2022a;b).

**Noise schedule:** Noise schedules significantly affect the performance of diffusion models (Chen, 2023). Many studies have proposed noise schedules (Ho et al., 2020; Nichol & Dhariwal, 2021; Chen, 2023), and Hoogeboom et al. (2023) proposed modifying noise schedules depending on the image resolution to be generated. However, the functional form in conventional studies, such as linear (Song et al., 2021a) and cosine (Nichol & Dhariwal, 2021), is somewhat arbitrary, and it remains unclear whether they are appropriately set. The most relevant work to ours is AYS (Sabour et al., 2024), which determines a noise schedule to minimize the KL divergence between the distributions of real data and generated samples. Although AYS enables us to obtain noise schedules tailored to the target dataset, sampler, NFE, and trained diffusion model, there are two shortcomings. First, AYS cannot be used as a noise schedule for training because a trained diffusion model is required to optimize noise schedules. Second, AYS is not applicable to several samplers, such as the deterministic sampler proposed in EDM (Karras et al., 2022).

In our work, we propose an efficient noise schedule for both training and sampling, which is applicable to any sampler.

## 3 BACKGROUND

In this section, we briefly introduce diffusion models.

Ho et al. (2020) proposed a discrete-time diffusion model. The forward process gradually adds Gaussian noise to the data $\boldsymbol{x}_0 \sim q(\boldsymbol{x}_0)$:

$$q(\boldsymbol{x}_{0:T}) = q(\boldsymbol{x}_0) \prod_{t=1}^{T} q(\boldsymbol{x}_t|\boldsymbol{x}_{t-1}), \tag{1}$$

$$q(\boldsymbol{x}_t|\boldsymbol{x}_{t-1}) = \mathcal{N}(\boldsymbol{x}_t; \beta_t \boldsymbol{x}_{t-1}, \delta_t^2 \boldsymbol{I}), \tag{2}$$

where $T$ is the number of timesteps in the forward process, $\beta_t$ is the pre-defined decay rate of the data, and $\delta_t$ denotes the noise strength added to $\boldsymbol{x}_{t-1}$. It is straightforward to verify that $q(\boldsymbol{x}_t|\boldsymbol{x}_0)$ is given by the following expression:

$$q(\boldsymbol{x}_t|\boldsymbol{x}_0) = \mathcal{N}(\boldsymbol{x}_t; \alpha_t \boldsymbol{x}_0, \sigma_t^2 \boldsymbol{I}), \tag{3}$$

where $\alpha_t = \beta_t \alpha_{t-1}$ and $\sigma_t^2 = \delta_t^2 + \beta_t^2 \sigma_{t-1}^2$. The aforementioned expression enables one-step sampling of $\boldsymbol{x}_t$ in the forward process. In this paper, we use $(\alpha_t, \sigma_t)$ rather than $(\beta_t, \delta_t)$ to characterize the forward process and adopt a variance-preserving process that satisfies $\alpha_t^2 + \sigma_t^2 = 1$.

The reverse process is also defined as a Markov chain, which starts from Gaussian noise:

$$p_\theta(\boldsymbol{x}_{0:T}) = p(\boldsymbol{x}_T) \prod_{t=1}^{T} p_\theta(\boldsymbol{x}_{t-1}|\boldsymbol{x}_t), \tag{4}$$

$$p(\boldsymbol{x}_T) = \mathcal{N}(\boldsymbol{x}_T; \boldsymbol{0}, \boldsymbol{I}), \tag{5}$$

where $\theta$ represents the model parameters to be learned. When $T$ is sufficiently large, $p_\theta(\boldsymbol{x}_{t-1}|\boldsymbol{x}_t)$ can be approximated by a Gaussian distribution (Feller, 2015) as follows:

$$p_\theta(\boldsymbol{x}_{t-1}|\boldsymbol{x}_t) = \mathcal{N}(\boldsymbol{x}_{t-1}; \boldsymbol{\mu}_\theta(\boldsymbol{x}_t, t), \nu_t^2 \boldsymbol{I}). \tag{6}$$

Under the above approximation, Ho et al. (2020) trained the noise prediction model $\boldsymbol{\varepsilon}_\theta(\boldsymbol{x}, t)$:

$$\boldsymbol{\mu}_\theta(\boldsymbol{x}, t) = \frac{1}{\beta_t} \left( \boldsymbol{x}_t - \frac{\delta_t^2}{\sigma_t} \boldsymbol{\varepsilon}_\theta(\boldsymbol{x}_t, t) \right), \tag{7}$$

rather than $\boldsymbol{\mu}_\theta(\boldsymbol{x}_t, t)$ to maximize a simplified version of the variational lower bound.

From the perspective of score-based matching, Song et al. (2021b) developed continuous-time diffusion models. In the continuous-time diffusion models, the forward and reverse processes are defined using the SDE and ODE. The reverse SDE and ODE of the variance-preserving process are given by (see Appendix C for the derivation):

$$\text{SDE} : d\boldsymbol{x} = \left( \boldsymbol{x} - \frac{2}{\sigma} \varepsilon_\theta(\boldsymbol{x}, \alpha) \right) \frac{d\alpha}{\alpha} + \sqrt{-2\frac{\dot{\alpha}}{\alpha}} d\boldsymbol{\omega}, \tag{8}$$

$$\text{ODE} : d\boldsymbol{x} = \left( \boldsymbol{x} - \frac{1}{\sigma} \boldsymbol{\varepsilon}_\theta(\boldsymbol{x}, \alpha) \right) \frac{d\alpha}{\alpha}. \tag{9}$$

$$\tag{10}$$

Recently, many samplers have been proposed to solve the above differential equations efficiently (Jolicoeur-Martineau et al., 2021; Liu et al., 2022; Lu et al., 2022a;b).

## 4 PROPOSED NOISE SCHEDULE

In this section, we explain our motivation and present our noise schedule.

### 4.1 MOTIVATION

As stated in Section 3, it is a good approximation to assume $p_\theta(\boldsymbol{x}_{t-1}|\boldsymbol{x}_t)$ as a Gaussian distribution if $T$ is sufficiently large. Note that $\Delta\alpha_t \equiv \alpha_t - \alpha_{t-1}$ is sufficiently small when $T \gg 1$. In principle, when refining the noise schedule, one should consider maximizing $\Delta\alpha_t$ as much as possible without breaking the aforementioned approximation. It is also conjectured that the threshold of $\Delta\alpha_t$, where the above approximation begins to break down, will vary depending on $\alpha_t$. (Otherwise, the noise schedule that linearly decreases $\alpha$ would always be the most efficient, but experimental results thus far do not support this.) Therefore, we need to appropriately control $\Delta\alpha_t$ when reducing the number of timesteps.

Song & Ermon (2020) proposed an alternative approach from the perspective of score matching. They considered an extremely simple case in which the dataset contains only one data point. The noise schedule is then determined so that consecutive probability distributions in the forward process have consistent overlap. Although this simple setting enables the analytical derivation of the noise schedule, the probability distribution of the target dataset is not considered.

We extend Song & Ermon (2020)'s idea and develop a generalized noise schedule. Similar to the score-based generative model (Song & Ermon, 2019) inspired by simulated annealing (Kirkpatrick et al., 1983), our formulation draws inspiration from the adiabatic theorem (Morita & Nishimori, 2008) in quantum annealing (Kadowaki & Nishimori, 1998). The adiabatic theorem formulates the traceability of the instantaneous probability distribution (ground state), and a noise schedule is devised on the basis of this traceability (Roland & Cerf, 2002). In quantum annealing, it is well known that the noise level must be changed slowly near phase-transition points, where macroscopic statistical quantities undergo significant changes.

## 4.2 Constant Rate Schedule

To ensure a consistent traceability throughout the diffusion process, our criterion to determine the noise schedule is to minimize the maximum distance between consecutive probability distributions in the forward process. Using the distance $D(t, t + \Delta t)$ between the probability distributions $p(\boldsymbol{x}_t)$ and $p(\boldsymbol{x}_{t+\Delta t})$, the aforementioned criterion is expressed as follows:

$$\min_{\alpha(t)} \left( \max_t D(t, t + \Delta t) \right). \tag{11}$$

The optimal noise schedule derived from the aforementioned criterion satisfies

$$D(t, t + \Delta t) = \frac{D(t, t + \Delta t) - D(t, t)}{\Delta t} \Delta t \simeq \left. \frac{\partial D(t, t')}{\partial t'} \right|_{t'=t} \Delta t = \text{const.}, \tag{12}$$

where we used $D(t, t) = 0$. Various methods exist for measuring the distance between two probability distributions, such as Kullback-Leibler divergence and Bhattacharyya distance. Song & Ermon (2020) approximated the probability distribution of the forward process by $p(\boldsymbol{x}_t) = \mathcal{N}(\boldsymbol{x}_t | \boldsymbol{0}, \sigma_t^2 \boldsymbol{I})$, and the distance is calculated using the overlap between $p(\boldsymbol{x}_t)$ and $p(\boldsymbol{x}_{t+\Delta t})$. Note that CRS is applicable to any distance metric.

To explicitly show the dependence on the noise schedule, we rewrite $D(t, t')$ as $\tilde{D}(\alpha(t), \alpha(t')) \equiv \tilde{D}(\alpha, \alpha')$. Here, $\alpha(t)$ is the noise-schedule function to be optimized. This results in

$$\left. \frac{\partial D(t, t')}{\partial t'} \right|_{t'=t} \Delta t = v(\alpha) \frac{d\alpha(t)}{dt} \Delta t = \text{const.}, \tag{13}$$

where

$$v(\alpha) = \left. \frac{\partial \tilde{D}(\alpha, \alpha')}{\partial \alpha'} \right|_{\alpha'=\alpha}. \tag{14}$$

Therefore, the noise-schedule function is determined to satisfy

$$\frac{d\alpha(t)}{dt} \propto v(\alpha)^{-1}. \tag{15}$$

The proportionality coefficient can be calculated using the boundary conditions: $\alpha(0) = 1$ and $\alpha(1) = 0$. Finally, we obtain the noise-schedule function as follows:

$$\frac{d\alpha(t)}{dt} = C v(\alpha)^{-\xi}, \tag{16}$$

$$C = \int_0^1 v(\alpha)^\xi d\alpha, \tag{17}$$

where we introduce a hyperparameter $\xi$ to enhance the flexibility of $\alpha(t)$.

## 4.3 Distance Metrics

We can use arbitrary distance metrics to obtain CRS. However, due to the high-dimensional nature of images, it is empirically known that accurately evaluating the probability distributions in pixel space is challenging (Lopez-Paz & Oquab, 2017). In the field of image generation, one of the most widely used metrics is the Fréchet Inception Distance (FID) (Heusel et al., 2017). The FID calculates the Fréchet distance under a normality assumption in the vision-relevant feature space embedded by the Inception-V3 model, where the dimension of the feature space is much smaller than that of pixel space. In this study, as an initial attempt, we adopt the FID as a distance metric for computing $\tilde{D}(\alpha, \alpha')$. We should note that the FID is not necessarily the optimal distance metric for CRS, and it remains as future work to identify the most suitable one.

In summary, we show the procedure to obtain CRS.

1. Chose a metric to measure the distance between probability distributions (FID in this paper).

2. Simulate the forward process and calculate $v(\alpha) \simeq \frac{\tilde{D}(\alpha, \alpha + \Delta \alpha)}{\Delta \alpha}$. See Appendix B for details.

3. Numerically integrate Eq. (16) using $v(\alpha)$ calculated in the previous step.

### 4.4 NUMERICAL INSTABILITY SUPPRESSION FOR SAMPLING

Up to this point, we have focused on changes in probability distributions. However, in this subsection, we introduce a technique to prevent numerical instability during sampling. As shown in Eqs. (8) and (9), the differential equations of the reverse process include $\frac{d\alpha}{\alpha}$. This term causes numerical instability when $\alpha \simeq 0$. Therefore, when discretizing the noise schedule for sampling, we impose the following constraint:

$$\frac{\Delta\alpha_t}{\alpha_t} \equiv \frac{\alpha_{t-1} - \alpha_t}{\alpha_t} \leq R, \tag{18}$$

where $R$ is a hyperparameter to control numerical instability. The pseudocode for calculating CRS for sampling is provided in Algorithm 1 in Appendix D.4.

## 5 EXPERIMENTS

We conduct experiments on unconditional and class-conditional image generation. We use six image datasets: LSUN (church/bedroom/horse/cat) (Yu et al., 2016), ImageNet (Deng et al., 2009), and FFHQ (Karras et al., 2019).

**Model $\varepsilon_\theta(\boldsymbol{x}, \alpha)$:** We use the U-Net model proposed in the ablated diffusion model (ADM) (Dhariwal & Nichol, 2021). The hyperparameters are shown in Table 7 in Appendix D.1.

**Noise schedule for training:** We use linear (Ho et al., 2020), cosine (Nichol & Dhariwal, 2021), shifted cosine (Hoogeboom et al., 2023), EDM training (Karras et al., 2022), and CRS. We minimize the following loss function:

$$L = \frac{1}{2}\mathbb{E}_{t \sim \mathcal{U}(0,1), \varepsilon \sim \mathcal{N}(\mathbf{0},\boldsymbol{I})} \left[ w(\alpha(t)) \left\| \varepsilon_\theta(\boldsymbol{x}, \alpha(t)) - \varepsilon \right\|^2 \right]. \tag{19}$$

We set $w(\alpha) = \alpha^{-2} - 0.75$ for EDM training (Kingma & Gao, 2023b), and $w(\alpha) = 1$ for other training schedules (simplified loss). See Appendix D.3 for details.

**Noise schedule for sampling:** We use linear (Ho et al., 2020), cosine (Nichol & Dhariwal, 2021), shifted cosine (Hoogeboom et al., 2023), EDM sampling (Karras et al., 2022), and CRS. Recent models use different noise schedules for training and sampling (Karras et al., 2022; Kingma & Gao, 2023a). Therefore, we evaluate the performance for arbitrary combinations of the training and sampling schedules. See Appendix D.4 for details.

**Sampling methods:** Two stochastic samplers and three deterministic samplers are adopted. Stochastic DDIM (DDIM, $\eta = 1$) (Song et al., 2021a) and SDE-DPM-Solver++(2M) (Lu et al., 2022b) are used as stochastic samplers. DDIM (Song et al., 2021a), PNDM (Liu et al., 2022), and DPM-Solver++(2M) (Lu et al., 2022b) are used as deterministic samplers.

**Evaluation metrics:** FID (Heusel et al., 2017), sFID (Nash et al., 2021), and improved precision and recall (Sajjadi et al., 2018; Kynkäänniemi et al., 2019) are evaluated. FID, sFID, and precision evaluate the sample fidelity, while recall evaluates the mode coverage. We generate 50K samples to calculate these metrics. All images in the training set are used to obtain the reference statistics for FID and sFID.

### 5.1 RESULTS ON LATENT-SPACE DIFFUSION MODELS

We trained the continuous-time diffusion models on LSUN (church/bedroom) for unconditional generation and ImageNet for class-conditional generation. The resolution is set to $256 \times 256$, and VQ-4 (Rombach et al., 2022) is used as the autoencoder, which is available in the GitHub repository: `https://github.com/CompVis/latent-diffusion`.

**Noise schedule:** To obtain CRS, we calculate $\tilde{D}(\alpha, \alpha')$ using the FID as the distance metric. The FID requires a feature model, which is trained in the latent space embedded by the autoencoder. We trained ResNet50 on ImageNet classification in the latent space and use it as the feature model. The noise schedules for sampling are shown in Figure 2. CRS varies depending on the dataset and significantly differs from conventional noise schedules. A common configuration of hyperparameters ($\xi = 1.0, R = 1.0, \alpha_{\min} = 0.01$) is used across all combinations of datasets, samplers, and NFEs,

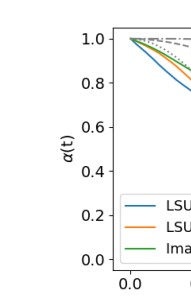 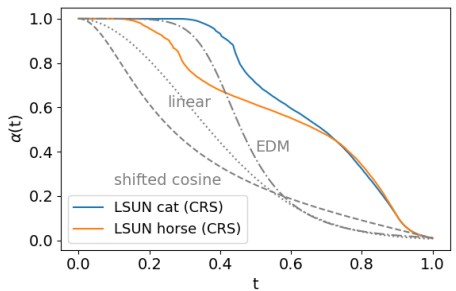

Figure 2: Comparison of noise schedules for sampling in latent-space diffusion models at NFE=100. CRS depends on the target datasets and significantly differs from conventional schedules.

Figure 3: Comparison of noise schedules for sampling in pixel-space diffusion models at NFE=250. A significant number of timesteps are consumed in weak-noise regions, which shows different trends from CRS in the latent-space diffusion models.

Table 1: Dependence on training and sampling noise schedules for LSUN church $256 \times 256$ in latent-space diffusion models.

| model | sampler | NFE | training schedule | sampling schedule | FID ↓ | precision ↑ | recall ↑ |
|-------|---------|-----|-------------------|-------------------|-------|-------------|----------|
| LDM | DDIM | 200 | linear | linear | 4.02 | 0.64 | 0.52 |
| Ours | DDIM | 100 | linear | linear | 4.17 | **0.61** | 0.55 |
| | | | cosine | | 4.56 | 0.60 | 0.56 |
| | | | EDM | | 5.11 | 0.54 | **0.57** |
| | | | CRS | | **4.16** | **0.61** | **0.57** |
| | | | CRS | linear | 4.16 | **0.61** | 0.57 |
| | | | | cosine | 17.35 | 0.53 | 0.51 |
| | | | | EDM | 4.70 | 0.60 | 0.54 |
| | | | | CRS | **3.79** | 0.60 | **0.58** |
| | SDE-DPM-Solver++(2M) | 100 | linear | linear | 3.51 | 0.64 | **0.57** |
| | | | cosine | | 3.87 | 0.64 | 0.56 |
| | | | EDM | | 3.70 | 0.64 | 0.53 |
| | | | CRS | | **3.44** | **0.65** | **0.57** |
| | | | CRS | linear | 3.44 | 0.65 | 0.57 |
| | | | | cosine | 3.53 | **0.66** | 0.55 |
| | | | | EDM | 3.43 | 0.64 | 0.56 |
| | | | | CRS | **3.24** | 0.64 | **0.58** |

except for $\alpha_{\min} = 0.001$ for ImageNet (see details in Table 12 in Appendix D.4). Here, $\alpha_{\min}$ is the minimum value of $\alpha$, at which the reverse process starts.

**LSUN church $256 \times 256$:** The results for LSUN church $256 \times 256$ are listed in Table 1. The bold numbers indicate the best training or sampling schedules. The rows shaded in gray represent the dependence of each metric on the training schedules. All metrics achieve their best values when CRS is used as the training schedule. The unshaded rows represent the dependence on the sampling schedule. FID and recall are further improved by using CRS as the sampling schedule. All results for other samplers are in Tables 13, 15, and 16

**LSUN bedroom $256 \times 256$:** The dependence on the sampling schedules is shown in Table 2. The training schedule is fixed to CRS. FID, precision, and recall achieve their best values when CRS is used as the sampling schedule. All results for other samplers and NFEs are in Tables 17 and 18.

**ImageNet $256 \times 256$:** Table 3 shows the results for class-conditional generation on ImageNet 256 $\times$ 256. The rows shaded in gray represent the dependence on the training schedules. FID, precision, and recall achieve their best values when CRS is used as the training schedule. On the other hand, as shown in the unshaded rows, the dependence on the sampling schedule is weaker compared with

Table 2: Dependence on sampling noise schedules for LSUN bedroom $256 \times 256$ in latent-space diffusion models.

| model | sampler | NFE | training schedule | sampling schedule | FID ↓ | precision ↑ | recall ↑ |
|-------|---------|-----|-------------------|-------------------|-------|-------------|----------|
| LDM | Stochastic DDIM | 200 | linear | linear | 2.95 | 0.66 | 0.48 |
| Ours | Stochastic DDIM | 100 | CRS | linear | 2.78 | 0.58 | 0.51 |
| | | | | cosine | 2.77 | 0.57 | 0.52 |
| | | | | EDM | 3.64 | 0.58 | 0.48 |
| | | | | CRS | **2.45** | **0.59** | **0.53** |
| | | 50 | CRS | linear | 4.14 | 0.57 | 0.47 |
| | | | | cosine | 4.31 | 0.56 | 0.48 |
| | | | | EDM | 7.91 | 0.52 | 0.39 |
| | | | | CRS | **3.28** | **0.58** | **0.49** |
| | SDE-DPM-Solver++(2M) | 100 | CRS | linear | 2.45 | **0.57** | 0.54 |
| | | | | cosine | 2.42 | 0.56 | 0.54 |
| | | | | EDM | 2.29 | **0.57** | 0.54 |
| | | | | CRS | **2.22** | **0.57** | **0.55** |
| | | 50 | CRS | linear | 2.44 | **0.57** | 0.53 |
| | | | | cosine | 2.84 | 0.55 | 0.54 |
| | | | | EDM | 2.49 | **0.57** | 0.54 |
| | | | | CRS | **2.33** | **0.57** | **0.55** |

Table 3: Dependence on training and sampling noise schedules for ImageNet $256 \times 256$ in latent-space diffusion models.

| model | sampler | NFE | training schedule | sampling schedule | FID ↓ | precision ↑ | recall ↑ |
|-------|---------|-----|-------------------|-------------------|-------|-------------|----------|
| LDM | DDIM | 250 | linear | linear | 10.56 | 0.71 | 0.62 |
| Ours | DDIM | 100 | linear | linear | 13.98 | **0.59** | 0.64 |
| | | | cosine | | 15.18 | 0.57 | **0.65** |
| | | | EDM | | 20.14 | 0.53 | **0.65** |
| | | | CRS | | **11.91** | **0.59** | **0.65** |
| | | | CRS | linear | 11.91 | **0.59** | 0.65 |
| | | | | cosine | 44.94 | 0.32 | **0.67** |
| | | | | EDM | 12.73 | **0.59** | 0.65 |
| | | | | CRS | **11.72** | **0.59** | 0.65 |
| | SDE-DPM-Solver++(2M) | 100 | linear | linear | 10.67 | **0.63** | 0.63 |
| | | | cosine | | 11.47 | 0.62 | 0.64 |
| | | | EDN | | 16.63 | 0.59 | 0.63 |
| | | | CRS | | **9.21** | **0.63** | **0.65** |
| | | | CRS | linear | 9.21 | 0.63 | **0.65** |
| | | | | cosine | 9.73 | 0.62 | 0.63 |
| | | | | EDM | **8.93** | **0.64** | 0.64 |
| | | | | CRS | 9.02 | **0.64** | 0.64 |

LSUN church and bedroom. Both the linear schedule and CRS tend to consistently work well across a wide range of samplers and NFEs. All results for other samplers are listed in Tables 14, 19, and 20.

## 5.2 RESULTS ON PIXEL-SPACE DIFFUSION MODELS

We trained the pixel-space diffusion models for unconditional image generation on LSUN (horse/cat). The resolution was set to $256 \times 256$. In the pixel-space diffusion models, we did not observe the superiority of CRS as the training schedule. Therefore, we evaluated the performance of CRS as the sampling schedule.

Table 4: Dependence on sampling noise schedules for LSUN horse $256 \times 256$ in pixel-space diffusion models.

| model | sampler | NFE | sampling schedule | metrics FID ↓ | sFID ↓ | precision ↑ | recall ↑ |
|-------|---------|-----|-------------------|-------|--------|-------------|----------|
| ADM | ddpm | 1000 | linear | 2.57 | 6.81 | 0.71 | 0.55 |
| Ours | SDE-DPM-Solver++(2M) | 250 | linear | 2.86 | 6.60 | 0.66 | **0.56** |
| | | | cosine | 3.09 | 7.13 | **0.68** | **0.56** |
| | | | EDM | 2.34 | 6.47 | **0.68** | **0.56** |
| | | | CRS | **2.30** | **6.34** | **0.68** | **0.56** |
| | DPM-Solver++(2M) | 250 | linear | 3.06 | 5.96 | 0.60 | **0.60** |
| | | | cosine | 2.72 | 5.58 | 0.62 | **0.60** |
| | | | EDM | 2.83 | 5.52 | 0.63 | 0.58 |
| | | | CRS | **2.68** | **5.36** | **0.64** | 0.59 |

Table 5: Dependence on sampling noise schedules for LSUN cat $256 \times 256$ in pixel-space diffusion models.

| model | sampler | NFE | sampling schedule | metrics FID ↓ | sFID ↓ | precision ↑ | recall ↑ |
|-------|---------|-----|-------------------|-------|--------|-------------|----------|
| ADM | ddpm | 1000 | linear | 5.57 | 6.69 | 0.63 | 0.52 |
| Ours | SDE-DPM-Solver++(2M) | 250 | linear | 6.35 | 7.20 | 0.59 | 0.49 |
| | | | cosine | 7.92 | 7.73 | 0.56 | 0.52 |
| | | | EDM | 5.26 | 7.09 | **0.60** | **0.53** |
| | | | CRS | **5.25** | **6.48** | **0.60** | **0.53** |
| | DPM-Solver++(2M) | 250 | linear | 6.22 | 6.33 | 0.50 | **0.57** |
| | | | cosine | 5.83 | 5.81 | 0.53 | **0.57** |
| | | | EDM | 5.89 | 5.99 | 0.54 | 0.56 |
| | | | CRS | **5.58** | **5.64** | **0.55** | 0.56 |

**Noise schedule:** We used the Inception-V3 model, trained on ImageNet classification, as the feature model for computing $\tilde{D}(\alpha, \alpha')$. CRS for each dataset is illustrated in Figure 3. Unlike in the latent-space diffusion models, many timesteps are commonly consumed in regions with low-noise levels. A common configuration of hyperparameters ($\xi = 1.2, R = 0.1, \alpha_{\min} = 0.01$) is used across all combinations of datasets, samplers, and NFEs (see details in Table 12 in Appendix D.4).

**LSUN horse $256 \times 256$ / cat $256 \times 256$:** The dependence of metrics on the sampling schedules is listed in Tables 4 and 5. The bold numbers indicate the best sampling schedules. CRS demonstrates the highest performance, except for recall with DPM-Solver++(2M). The evaluation results for other samplers are listed in Tables 21 and 22.

## 5.3 ADDITIONAL EVALUATION ON EDM

We additionally evaluate the superiority of CRS as the sampling schedule using the pre-trained models in EDM (Karras et al., 2022).

**FFHQ $64 \times 64$:** We evaluated FID using three sampling schedules, as shown in Table 6. AYS (Sabour et al., 2024) is a recently proposed noise schedule, that systematically optimizes the sampling schedule with respect to the trained diffusion models, datasets, samplers, and NFEs. In this experiment, the hyperparameters of CRS are tuned for each combination of samplers and NFEs, as is done in AYS (see details in Table 12 in Appendix D.4). CRS consistently outperforms the sampling schedule of EDM and exhibits better FID than AYS when DPM-Solver++(2M) is used. The underlined values indicate the best FID across all combinations of sampling schedules and samplers for each NFE. The best FID is achieved when using CRS for all NFEs. Here, we should note that AYS is not applicable to the deterministic sampler proposed in EDM (Deterministic EDM), though the best FID is obtained using the Deterministic EDM for NFE = 30 and 50.

Table 6: Dependence on sampling noise schedules for FFHQ $64 \times 64$ in EDM. The underlined values indicate the best FID across all combinations of sampling schedules and samplers for each NFE.

| sampler | sampling schedule | NFE=20 | NFE=30 | NFE=50 |
|---|---|---|---|---|
| Deterministic EDM | EDM | 4.34 | 2.88 | 2.52 |
| | CRS | **3.49** | **2.57** | **2.33** |
| DPM-Solver++(2M) | EDM | 3.12 | 2.72 | **2.54** |
| | AYS (Sabour et al., 2024) | 3.29 | 2.87 | 2.62 |
| | CRS | **3.08** | **2.66** | **2.54** |
| SDE-DPM-Solver++(2M) | EDM (Sabour et al., 2024) | 9.67 | 5.96 | 3.85 |
| | AYS (Sabour et al., 2024) | **5.65** | **3.97** | **3.13** |
| | CRS | 7.57 | 4.97 | 3.47 |

## 6 Discussion

Through extensive experiments, we confirmed that CRS broadly improves the performance of diffusion models regardless of the dataset, sampler, NFE, or type of diffusion model. However, CRS has room for improvement in the selection of the distance metric. In this work, we adopt FID as the distance metric, which requires a feature model. We acknowledge that the feature models used in this work have the following two issues. The first is that, since the feature models are trained on ImageNet classification, there are concerns about whether they can accurately capture the features of the target datasets. The second is that, as the feature models have not been exposed to diffused images during training, they cannot correctly capture the changes in the probability distributions in the forward process. We expect that self-supervised learning is one of the candidate approaches for obtaining better feature models. Exploration of feature models and other distance metrics remains as future work. By using a more appropriate distance metric, CRS might also be applicable as a training schedule in pixel-space diffusion models.

CRS in the latent-space diffusion model significantly differs from conventional noise schedules. The most remarkable difference is the rate of change in $\alpha(t)$ within low-noise regions. In pixel-space diffusion models, a significant number of timesteps are consumed in regions with low noise. It is presumed that, in the pixel-space diffusion models, perceptual expressions are generated in these low-noise regions. On the other hand, in latent-space diffusion models, the autoencoder is responsible for generating perceptual expressions, indicating that there is no need to allocate excessive resources to regions with low-noise levels.

## 7 Conclusion

We proposed CRS, which ensures a constant rate of change in the probability distribution throughout the diffusion process. CRS does not require a pre-defined functional form of the noise schedule; instead, the functional form tailored to the target dataset and type of diffusion model is systematically derived. In addition, CRS can be applied not only to the sampling schedule combined with arbitrary samplers but also to the training schedule. Through extensive experiments, we confirmed that CRS broadly improves the performance of diffusion models. The most significant challenge with CRS is establishing a method for measuring the distance between probability distributions in the forward process. By identifying a more appropriate distance metric, greater improvements may be achieved. Furthermore, CRS is applicable beyond image generation, such as audio and 3D-point cloud generation, if we can appropriately define the distance. We aim to address this in future work.

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

## A  PROBABILITY DISTRIBUTION OF DIFFUSED DATA

We derive the probability distribution of the diffused data used in the toy example (Figure 1).

The probability distribution of the diffused data is given by

$$q(\boldsymbol{x}_\alpha) = \int q(\boldsymbol{x}_\alpha|\boldsymbol{x}_0)q(\boldsymbol{x}_0), \tag{20}$$

$$q(\boldsymbol{x}_\alpha|\boldsymbol{x}_0) = \mathcal{N}(\boldsymbol{x}_\alpha; \alpha\boldsymbol{x}_0, \sigma^2\boldsymbol{I}), \tag{21}$$

where $q(\boldsymbol{x}_0)$ is the probability distribution of a target dataset. We then approximate $q(\boldsymbol{x}_0)$ with the empirical distribution as follows:

$$q(\boldsymbol{x}_0) = \frac{1}{N}\sum_{n=1}^{N}\delta(\boldsymbol{x}_0 - \boldsymbol{x}_0^n), \tag{22}$$

where $N$ is the number of samples in the target dataset, $\boldsymbol{x}_0^n$ is the $n$-th sample, and $\delta$ denotes Dirac's delta function. By substituting Eq. (22) into Eq. (20), we obtain

$$q(\boldsymbol{x}_\alpha) = \frac{1}{N}\sum_{n=1}^{N}\mathcal{N}(\boldsymbol{x}_\alpha; \alpha\boldsymbol{x}_0^n, \sigma^2\boldsymbol{I}). \tag{23}$$

Here, we used the following formula of Dirac's delta function:

$$\int f(\boldsymbol{x})\delta(\boldsymbol{x} - \boldsymbol{y})d\boldsymbol{x} = f(\boldsymbol{y}). \tag{24}$$

The resulting probability distribution is a Gaussian mixture distribution. We used Eq. (23) in the toy example.

## B  CALCULATION OF DISTANCE

We provide details on how to calculate $\tilde{D}(\alpha, \alpha')$ on the basis of the FID.

For each training image $\boldsymbol{x}^{(n)}$, we generate a trajectory $\{\boldsymbol{x}_t^{(n)}|t = 0, 1, ..., T\}$ using the following update rule of the forward process:

$$\boldsymbol{x}_t^{(n)} = \beta_t\boldsymbol{x}_{t-1}^{(n)} + \delta_t\boldsymbol{\varepsilon}, \tag{25}$$

$$\boldsymbol{\varepsilon} \sim \mathcal{N}(\boldsymbol{0}, \boldsymbol{I}), \tag{26}$$

where $N$ represents the number of training data. The next step is to calculate the mean and covariance matrix in the feature space as follows:

$$\boldsymbol{\mu}_t = \frac{1}{N}\sum_{n=1}^{N}\boldsymbol{\phi}(\boldsymbol{x}_t^{(n)}), \tag{27}$$

$$\boldsymbol{\Sigma}_t = \frac{1}{N-1}\left(\sum_{n=1}^{N}\boldsymbol{\phi}(\boldsymbol{x}_t^{(n)})\boldsymbol{\phi}^T(\boldsymbol{x}_t^{(n)}) - N\boldsymbol{\mu}_t\boldsymbol{\mu}_t^T\right). \tag{28}$$

Here, $\boldsymbol{\phi}$ is the feature model. Using $\boldsymbol{\mu}_t$ and $\boldsymbol{\Sigma}_t$, $D(t, t')$ is obtained as

$$D(t, t') = \|\boldsymbol{\mu}_t - \boldsymbol{\mu}_{t'}\|_2^2 + \text{Tr}\left(\boldsymbol{\Sigma}_t + \boldsymbol{\Sigma}_{t'} - 2\sqrt{\boldsymbol{\Sigma}_t\boldsymbol{\Sigma}_{t'}}\right). \tag{29}$$

If the noise-schedule function $\alpha(t)$ is monotonically decreasing, an inverse function $\alpha^{-1}(\alpha_t) = t$ exists. Using the inverse function, we obtain $\tilde{D}(\alpha, \alpha')$ as follows:

$$\tilde{D}(\alpha, \alpha') = D(\alpha^{-1}(\alpha_t), \alpha^{-1}(\alpha_{t'})). \tag{30}$$

Here, we should note that $\{\alpha_t|t = 0, 1, ..., T\}$ must be pre-defined before calculating $\tilde{D}(\alpha, \alpha')$. The pre-defined noise schedule can be considered as an initial value and is updated with our algorithm. Depending on the initial value, it may be necessary to carry out multiple updates.

## C  DERIVATION OF SDE AND ODE FOR THE REVERSE PROCESS

The forward SDE of the score-based model (Song et al., 2021b) is defined as

$$d\boldsymbol{x} = \boldsymbol{f}(\boldsymbol{x}, t)dt + g(t)d\boldsymbol{\omega}, \tag{31}$$

where $\boldsymbol{\omega}$ is the standard Wiener process, $\boldsymbol{f}(\boldsymbol{x}, t)$ is the drift coefficient, and $g(t)$ is the diffusion coefficient. For the variance-preserving and variance-exploding SDEs, $\boldsymbol{f}(\boldsymbol{x}, t)$ is given in a separable form: $\boldsymbol{f}(\boldsymbol{x}, t) = f(t)\boldsymbol{x}$. In this case, the reverse SDE and ODE are given by

$$\text{SDE} : d\boldsymbol{x} = \left[ f(t)\boldsymbol{x} - g(t)^2 \nabla_{\boldsymbol{x}} \log p_t(\boldsymbol{x}) \right] dt + g(t)d\boldsymbol{\omega}, \tag{32}$$

$$\text{ODE} : d\boldsymbol{x} = \left[ f(t)\boldsymbol{x} - \frac{g(t)^2}{2} \nabla_{\boldsymbol{x}} \log p_t(\boldsymbol{x}) \right] dt, \tag{33}$$

where $p_t(\boldsymbol{x})$ represents the diffused data distribution.

The diffused data distributions conditioned on $\boldsymbol{x}(0)$ have the general form (Song et al., 2021b; Karras et al., 2022):

$$p(\boldsymbol{x}(t)|\boldsymbol{x}(0)) = \mathcal{N}(\boldsymbol{x}(t); \alpha(t)\boldsymbol{x}(0), \sigma(t)^2 \boldsymbol{I}), \tag{34}$$

where

$$\alpha(t) = \exp\left( \int_0^t f(\xi)d\xi \right), \tag{35}$$

and

$$\frac{\sigma(t)}{\alpha(t)} = \sqrt{\int_0^t \frac{g(\xi)^2}{s(\xi)^2}}. \tag{36}$$

Using Eqs. (35) and (36), $f(t)$ and $g(t)$ can be expressed in terms of $\alpha(t)$ and $\sigma(t)$ as follows:

$$f(t) = \frac{\dot{\alpha}(t)}{\alpha(t)}, \tag{37}$$

$$\frac{1}{2}g(t)^2 = \sigma(t)\dot{\sigma}(t) - \sigma(t)^2 \frac{\dot{\alpha}(t)}{\alpha(t)}, \tag{38}$$

where the dot denotes a time derivative. Thus, we can rewrite the reverse SDE and ODE using $\alpha(t)$ and $\sigma(t)$:

$$\text{SDE} : d\boldsymbol{x} = \left[ \frac{\dot{\alpha}(t)}{\alpha(t)}\boldsymbol{x} - 2\left( \sigma(t)\dot{\sigma}(t) - \sigma(t)^2 \frac{\dot{\alpha}(t)}{\alpha(t)} \right) \nabla_{\boldsymbol{x}} \log p_t(\boldsymbol{x}) \right] dt$$

$$+ \sqrt{2\left( \sigma(t)\dot{\sigma}(t) - \sigma(t)^2 \frac{\dot{\alpha}(t)}{\alpha(t)} \right)} d\boldsymbol{\omega}, \tag{39}$$

$$\text{ODE} : d\boldsymbol{x} = \left[ \frac{\dot{\alpha}(t)}{\alpha(t)}\boldsymbol{x} - \left( \sigma(t)\dot{\sigma}(t) - \sigma(t)^2 \frac{\dot{\alpha}(t)}{\alpha(t)} \right) \nabla_{\boldsymbol{x}} \log p_t(\boldsymbol{x}) \right] dt. \tag{40}$$

$$\tag{41}$$

In the variance-preserving process, the following two equations are satisfied:

$$\alpha(t)^2 + \sigma(t)^2 = 1, \tag{42}$$

$$\alpha(t)\dot{\alpha}(t) + \sigma(t)\dot{\sigma}(t) = 0. \tag{43}$$

Thus, we can rewrite the reverse SDE and ODE as follows:

$$\text{SDE} : d\boldsymbol{x} = \frac{\dot{\alpha}(t)}{\alpha(t)} \left[ \boldsymbol{x} + 2\nabla_{\boldsymbol{x}} \log p_t(\boldsymbol{x}) \right] dt + \sqrt{-2\frac{\dot{\alpha}(t)}{\alpha(t)}} d\boldsymbol{\omega}, \tag{44}$$

$$\text{ODE} : d\boldsymbol{x} = \frac{\dot{\alpha}(t)}{\alpha(t)} \left[ \boldsymbol{x} + \nabla_{\boldsymbol{x}} \log p_t(\boldsymbol{x}) \right] dt. \tag{45}$$

Finally, by substituting the following definition into Eqs. (44) and (45), we obtain Eqs. (8) and (9):

$$\nabla \log p_t(\boldsymbol{x}) = -\frac{\boldsymbol{\varepsilon}_\theta(\boldsymbol{x}, \alpha(t))}{\sigma(t)}. \tag{46}$$

Table 7: Hyperparameters of the U-Net model

|  | Latent-space diffusion models | Pixel-space diffusion models |
| --- | --- | --- |
| resolution | 64 | 256 |
| Number of parameters | 296M | 552M |
| Channels | 192 | 256 |
| Depth | 3 | 2 |
| Channels multiple | 1,2,3,4 | 1,1,2,2,4,4 |
| Heads Channels | 64 | 64 |
| Attention resolution | 32,16,8 | 32,16,8 |
| BigGAN up/downsample | True | True |
| Dropout | 0.1 | 0.1 |

Table 8: Hyperparameters for training the latent-space diffusion models.

|  | LSUN bedroom $256 \times 256$ | LSUN church $256 \times 256$ | ImageNet $256 \times 256$ |
| --- | --- | --- | --- |
| Epochs | 100 | 1000 | 500 |
| EMA decay rate | 0.9999 | 0.9999 | 0.9999 |
| Optimizer | Adam | Adam | Adam |
| $\beta_1$ | 0.9 | 0.9 | 0.9 |
| $\beta_2$ | 0.999 | 0.999 | 0.999 |
| Learning rate | 1e-4 | 1e-4 | 1e-4 |
| Batch size | 32x8 | 32x8 | 32x8 |
| Number of GPUs | A100x8 | A100x8 | A100x8 |
| Training time | 11 days | 5 days | 22 days |

# D IMPLEMENTATION DETAILS

## D.1 UNET MODEL

Hyperparameters of U-Net model are listed in Table 7. For latent-space diffusion models, we used the same values as the ImageNet $64 \times 64$ in ADM (Dhariwal & Nichol, 2021). In addition, for pixel-space diffusion models, we used the same values as the LSUN in ADM (Dhariwal & Nichol, 2021).

## D.2 TRAINING

Hyperparameters for training the diffusion models are summarized in Tables 8 and 9. We save 10 checkpoints for each model. The FID score with 10,000 samples is evaluated for all checkpoints, and the results are reported with the checkpoint with the best FID for each combination of the sampler and NFE.

Table 9: Hyperparameters for training the pixel-space diffusion models.

|  | LSUN cat $256 \times 256$ | LSUN horse $256 \times 256$ |
| --- | --- | --- |
| Epochs | 50 | 50 |
| EMA decay rate | 0.9999 | 0.9999 |
| Optimizer | Adam | Adam |
| $\beta_1$ | 0.9 | 0.9 |
| $\beta_2$ | 0.999 | 0.999 |
| Learning rate | 1e-4 | 1e-4 |
| Batch size | 32x8 | 32x8 |
| Number of GPUs | A100x8 | A100x8 |
| Training time | 11 days | 14 days |

Table 10: Settings of the loss function.

| setting name | training noise schedule: $\alpha(t)$ | weighting function: $w(\alpha)$ |
|---|---|---|
| linear | linearly interpolate the following points:
$\alpha_0 = 1$,
$\alpha_{1 \le i \le 1000} = \sqrt{1 - \tilde{\beta}_i}\alpha_{i-1}$,
$\tilde{\beta}_i = 10^{-4} + \frac{0.02 - 10^{-4}}{999}(i-1)$. | $w(\alpha) = 1$ |
| cosine | $\alpha(t) = \frac{f(t)}{f(0)}$,
$f(t) = \cos\left(\frac{t+0.008}{1.008}\frac{\pi t}{2}\right)$. | $w(\alpha) = 1$ |
| EDM | $\alpha(t) = \frac{1}{\sqrt{1+e^{-\lambda(t)}}}$,
$\lambda(t) = -F_{\mathcal{N}}^{-1}(t; 2.4, 2.4^2)$,
where $F_{\mathcal{N}}^{-1}$ is the cumulative density function
of the Normal distribution. | $w(\alpha) = \alpha^{-2} - 0.75$ |
| CRS | $\frac{d\alpha}{dt} \propto v(\alpha)^{-\xi}$,
where $\xi$ is set to 1. | $w(\alpha) = 1$ |

### D.3 LOSS FUNCTION

According to Kingma & Gao (2023b), the loss function of the diffusion models has the general form:

$$L = \frac{1}{2}\mathbb{E}_{t \sim \mathcal{U}(0,1), \varepsilon \sim \mathcal{N}(\mathbf{0}, \mathbf{I})}\left[\tilde{w}\big(\lambda(t)\big) \cdot -\frac{d\lambda(t)}{dt} \cdot \left\|\tilde{\varepsilon}_\theta\big(\boldsymbol{x}(t), \lambda(t)\big) - \boldsymbol{\varepsilon}\right\|^2\right], \tag{47}$$

where $\tilde{w}\big(\lambda(t)\big)$ is the weight for each noise level, and $\lambda(t)$ is the signal-to-noise ratio defined by

$$\lambda(t) = \log\left(\frac{\alpha(t)^2}{\lambda(t)^2}\right). \tag{48}$$

The aforementioned loss function with $\tilde{w}\big(\lambda(t)\big) = -\big(d\lambda(t)/dt\big)^{-1}$ is identical to the simplified loss function proposed in Ho et al. (2020). In addition, in EDM (Karras et al., 2022), $\tilde{w}\big(\lambda(t)\big)$ and $\lambda(t)$ are set as follows (Kingma & Gao, 2023b):

$$\tilde{w}\big(\lambda(t)\big) = \mathcal{N}\big(\lambda(t); 2.4, 2.4^2\big)(e^{-\lambda(t)} + 0.5^2), \tag{49}$$

$$-\frac{d\lambda(t)}{dt} = \mathcal{N}\big(\lambda(t); 2.4, 2.4^2\big)^{-1}. \tag{50}$$

In our experiments, we optimized the following loss function:

$$L = \frac{1}{2}\mathbb{E}_{t \sim \mathcal{U}(0,1), \varepsilon \sim \mathcal{N}(\mathbf{0}, \boldsymbol{I})}\left[w\big(\alpha(t)\big)\left\|\varepsilon_\theta\big(\boldsymbol{x}(t), \alpha(t)\big) - \boldsymbol{\varepsilon}\right\|^2\right], \tag{51}$$

where

$$w\big(\alpha(t)\big) = \tilde{w}\big(\lambda(t)\big) \cdot -\frac{d\lambda(t)}{dt}, \tag{52}$$

and

$$\varepsilon_\theta\big(\boldsymbol{x}(t), \alpha(t)\big) = \tilde{\varepsilon}_\theta\big(\boldsymbol{x}(t), \lambda(t)\big). \tag{53}$$

The training noise schedule and loss weights used in our experiments are summarized in Table 10.

### D.4 NOISE SCHEDULE FOR SAMPLING

Noise schedules for sampling evaluated in our work are summarized in Table 11. $T$ is the number of timesteps for sampling. Hyperparameters are listed in Table 12. A common configuration ($\alpha_{\min} = 0.01, \xi = 1.0, R = 1.0$) is used across all combinations of datasets, samplers, and NFEs in the latent-space diffusion models, except for $\alpha_{\min}$ on ImageNet $256 \times 256$. Similarly, a common configuration ($\alpha_{\min} = 0.01, \xi = 1.2, R = 0.1$) is used in the pixel-space diffusion models. On the other hand, the hyperparameters are optimized for each combination of sampler and NFE in EDM, as is done in AYS (Sabour et al., 2024).

Table 11: CRS for sampling.

| schedule name | noise schedule for sampling: $\{\alpha_i | i = 0, 1, ..., T\}$ |
|---|---|
| linear | $\alpha_i = \alpha\left(\frac{i}{T}\right)$, 
 where $\alpha(t)$ is the noise schedule for training. |
| cosine | $\alpha_0 = 1$, 
 $\alpha_{i+1} = \max\left(\sqrt{10^{-3}}\alpha_i, \alpha\left(\frac{i}{T}\right)\right)$, 
 where $\alpha(t)$ is the noise schedule for training. |
| shifted cosine | $\alpha_i = \alpha\left(\frac{i}{T}\right)$, 
 $\alpha(t) = (\alpha_{\max} - \alpha_{\min})f(t) + \alpha_{\min}$, 
 $f(t) = \cos\left(\frac{\pi t}{2}\right)\frac{r}{\sqrt{1+(r^2-1)\cos^2\left(\frac{\pi t}{2}\right)}}$, 
 $\alpha_{\max} = 1.0, \alpha_{\min} = 0.01, r = \frac{64}{s}$, 
 where $s$ is the resolution. |
| EDM | $\alpha_0 = 1$, 
 $\alpha_{i\geq 1} = \alpha\left(\frac{i}{T}\right)$, 
 $\alpha(t) = \frac{1}{\sqrt{1+e^{-\lambda(t)}}}$, 
 $\lambda(t) = \log\frac{1}{\sigma(t)^2}$, 
 $\sigma(t) = \left\{\sigma_{\max}^{\frac{1}{\rho}} + \left(\sigma_{\min}^{\frac{1}{\rho}} - \sigma_{\max}^{\frac{1}{\rho}}\right)(1-t)\right\}^{\rho}$, 
 $\sigma_{\max} = 80, \sigma_{\min} = 0.002, \rho = 7$. |
| CRS | see Algorithm 1 |

---

**Algorithm 1** CRS for sampling

---

**Require:** $\alpha_{\min}, R, \xi, T, v(\alpha)$  # T: number of timesteps for sampling
**Ensure:** $\{\alpha_i | i = 0, 1, ..., T\}$
1: $\alpha_0 \leftarrow 1, \ \alpha_T \leftarrow \alpha_{\min}$
2: $C \leftarrow \int_{\alpha_{\min}}^{1} v(\alpha)^\xi d\alpha$
3: obtain $\alpha(t)$ by solving Eq. (16)
4: **for** $i = T$ to 1 **do**
5: $\quad \alpha_{\text{cand}} \leftarrow \alpha\left(\frac{i-1}{T}\right)$
6: $\quad R_{\text{cand}} \leftarrow \frac{\alpha_{\text{cand}} - \alpha_i}{\alpha_i}$
7: $\quad$ **if** $R_{\text{cand}} < R$ **then**
8: $\quad\quad \alpha_{i-1} \leftarrow \alpha_{\text{cand}}$
9: $\quad$ **else**
10: $\quad\quad \alpha_{i-1} \leftarrow (1 + R)\alpha_i$
11: $\quad\quad C \leftarrow \int_{\alpha_{i-1}}^{1} v(\alpha)^\xi d\alpha$
12: $\quad\quad$ obtain $\alpha(t)$ by solving Eq. (16)
13: $\quad$ **end if**
14: **end for**

---

Table 12: Hyperparameters for sampling schedules.

| model | dataset | sampler | NFE | $\alpha_{\min}$ | $\xi$ | R |
|---|---|---|---|---|---|---|
| latent space | church 256 × 256 bedroom 256 × 256 | Stochastic DDIM SDE-DPM-Solver++(2M) DDIM PNDM DPM-Solver++(2M) | 100 50 30 | 0.01 | 1.0 | 1.0 |
| | ImageNet 256 × 256 | Stochastic DDIM SDE-DPM-Solver++(2M) DDIM PNDM DPM-Solver++(2M) | 100 50 30 | 0.001 | 1.0 | 1.0 |
| pixel space | cat 256 × 256 horse 256 × 256 | SDE-DPM-Solver++(2M) PNDM DPM-Solver++(2M) | 250 | 0.01 | 1.2 | 0.1 |
| EDM | FFHQ 64 × 64 | SDE-DPM-Solver++(2M) | 50 | 0.0125 | 1.3 | 0.2 |
| | | | 30 | 0.0125 | 1.4 | 0.3 |
| | | | 20 | 0.0125 | 1.3 | 0.7 |
| | | DPM-Solver++(2M) | 50 | 0.0125 | 1.5 | 0.2 |
| | | | 30 | 0.0125 | 1.5 | 0.3 |
| | | | 20 | 0.0125 | 1.5 | 0.5 |
| | | EDM deterministic | 50 | 0.0125 | 1.5 | 1.0 |
| | | | 30 | 0.0125 | 1.5 | 1.2 |
| | | | 20 | 0.0125 | 1.5 | 1.7 |

## E    DETAILED RESULTS

We list all evaluation results. The dependence on the training schedule is listed in the following tables:

- Latent-space diffusion model on LSUN church 256 × 256: Table 13.
- Latent-space diffusion model on ImageNet 256 × 256: Table 14.

The dependence on the sampling schedule is listed in the following tables:

- Latent-space diffusion model on LSUN church 256 × 256: Tables 15, and 16.
- Latent-space diffusion model on LSUN bedroom 256 × 256: Tables 17, and 18.
- Latent-space diffusion model on ImageNet 256 × 256: Tables 19, and 20.
- Pixel-space diffusion model on LSUN horse 256 × 256: Table 21.
- Pixel-space diffusion model on LSUN cat 256 × 256: Table 22.

## F    GENERATED SAMPLES

Images sampled from diffusion models are shown in the following figures:

- The latent-space diffusion model on LSUN church 256 × 256: Figure 4.
- The latent-space diffusion model on LSUN bedroom 256 × 256: Figure 5.
- The latent-space diffusion model on ImageNet 256 × 256: Figure 6, and 7.
- The pixel-space diffusion model on LSUN horse 256 × 256: Figure 8.
- The pixel-space diffusion model on LSUN cat 256 × 256: Figure 9.

Table 13: FID / recall evaluated on LSUN church $256 \times 256$ in the latent-space diffusion model with arbitrary combinations of the training and sampling schedules at NFE=100. The bold numbers indicate the best training schedule for each sampling schedule. The underlined values represent the optimal or equivalent combinations of the training and sampling schedules with respect to FID and recall. By using CRS for both the training and sampling schedules, high fidelity and mode coverage are achieved across all samplers.

| sampler | training schedule | sampling schedule linear | cosine | EDM | CRS |
|---|---|---|---|---|---|
| Stochastic DDIM | linear | 4.37 / **0.54** | 4.76 / 0.49 | 5.77 / 0.47 | **3.98 / 0.55** |
| | cosine | 4.84 / 0.51 | 6.37 / **0.52** | 6.46 / 0.48 | 4.23 / **0.55** |
| | EDM | **4.36** / 0.50 | 50.89 / 0.43 | **5.65** / 0.47 | 4.06 / 0.51 |
| | CRS | **4.36** / 0.53 | **4.49** / 0.52 | 5.78 / **0.49** | 4.01 / **0.55** |
| SDE-DPM-Solver++(2M) | linear | 3.51 / **0.57** | 3.71 / 0.53 | 3.48 / **0.57** | **3.22** / 0.58 |
| | cosine | 3.87 / 0.56 | 4.72 / **0.55** | 3.90 / 0.56 | 3.40 / **0.59** |
| | EDM | 3.70 / 0.53 | 39.10 / 0.46 | 3.68 / 0.53 | 3.48 / 0.54 |
| | CRS | **3.44** / 0.57 | **3.53** / 0.55 | **3.43** / 0.56 | 3.24 / 0.58 |
| DDIM | linear | 4.17 / 0.55 | 20.26 / 0.42 | 4.90 / 0.54 | 3.92 / 0.58 |
| | cosine | 4.56 / 0.56 | 45.77 / **0.51** | 5.33 / **0.55** | 4.11 / 0.58 |
| | EDM | 5.11 / **0.57** | 174.11 / 0.45 | 4.98 / **0.55** | 4.58 / **0.59** |
| | CRS | **4.16** / 0.57 | **17.35** / 0.51 | **4.70** / 0.54 | **3.79** / 0.58 |
| PNDM | linear | 3.73 / 0.57 | **218.84** / 0.00 | 3.77 / 0.58 | 3.57 / 0.59 |
| | cosine | 3.97 / 0.58 | 232.12 / 0.00 | 4.05 / **0.58** | 3.81 / **0.60** |
| | EDM | 4.75 / **0.59** | 249.75 / 0.00 | 4.22 / **0.58** | 4.25 / 0.59 |
| | CRS | **3.65** / 0.58 | 230.27 / 0.00 | **3.76** / **0.58** | **3.48** / 0.59 |
| DPM-Solver++(2M) | linear | 3.76 / 0.57 | 15.49 / 0.44 | **3.85** / 0.58 | 3.59 / 0.59 |
| | cosine | 4.01 / 0.58 | 37.34 / **0.53** | 4.15 / **0.58** | 3.79 / **0.60** |
| | EDM | 4.80 / **0.59** | 164.82 / 0.48 | 4.28 / **0.58** | 4.30 / 0.59 |
| | CRS | **3.70** / 0.58 | **13.88** / 0.52 | 3.86 / **0.58** | **3.47** / 0.59 |

Table 14: FID / recall evaluated on ImageNet $256 \times 256$ in the latent-space diffusion model with arbitrary combinations of the training and sampling schedules at NFE=100. The bold numbers indicate the best training schedule for each sampling schedule. The underlined values represent the optimal or equivalent combinations of the training and sampling schedules with respect to FID and recall. By using CRS for both the training and sampling schedules, high fidelity and mode coverage are achieved across all samplers at NFE=100.

| sampler | training schedule | sampling schedule linear | cosine | EDM | CRS |
|---|---|---|---|---|---|
| Stochastic DDIM | linear | 12.36 / **0.62** | 12.49 / 0.60 | 14.95 / 0.59 | 11.88 / 0.61 |
| | cosine | 13.23 / **0.62** | 16.53 / 0.62 | 16.08 / 0.60 | 12.59 / **0.62** |
| | EDM | 18.24 / 0.61 | 94.47 / **0.64** | 20.89 / 0.60 | 17.89 / 0.61 |
| | CRS | **10.98** / 0.62 | **12.13** / 0.62 | **13.33** / 0.61 | **10.15 / 0.62** |
| SDE-DPM-Solver++(2M) | linear | 10.67 / 0.63 | 10.74 / 0.62 | 10.58 / 0.63 | 10.84 / 0.63 |
| | cosine | 11.47 / 0.64 | 12.67 / 0.63 | 11.19 / 0.63 | 11.31 / **0.64** |
| | EDM | 16.63 / 0.63 | 70.62 / **0.65** | 16.58 / 0.63 | 16.95 / 0.63 |
| | CRS | **9.21 / 0.65** | **9.73** / 0.63 | **8.93** / **0.64** | **9.02** / 0.64 |
| DDIM | linear | 13.98 / 0.64 | **38.63** / 0.61 | 14.75 / 0.64 | 13.19 / 0.63 |
| | cosine | 15.18 / **0.65** | 75.51 / 0.66 | 16.08 / **0.65** | 15.79 / 0.64 |
| | EDM | 20.14 / **0.65** | 166.80 / **0.67** | 20.66 / **0.65** | 30.02 / **0.66** |
| | CRS | **11.91** / **0.65** | 44.94 / **0.67** | **12.73** / **0.65** | **11.72** / 0.65 |
| PNDM | linear | 13.24 / **0.65** | 201.58 / 0.00 | 13.40 / 0.64 | 12.77 / 0.64 |
| | cosine | 14.18 / **0.65** | 185.13 / 0.00 | 14.47 / 0.65 | 14.34 / **0.65** |
| | EDM | 19.69 / **0.65** | **170.84** / 0.00 | 20.24 / 0.65 | 22.86 / **0.65** |
| | CRS | **11.16** / **0.65** | 176.01 / 0.00 | **11.43** / **0.66** | **11.00 / 0.65** |
| DPM-Solver++(2M) | linear | 13.15 / **0.65** | **32.91** / 0.61 | 13.48 / **0.65** | 12.91 / 0.64 |
| | cosine | 14.30 / **0.65** | 66.27 / 0.66 | 14.58 / **0.65** | 14.55 / **0.65** |
| | EDM | 19.77 / **0.65** | 161.38 / **0.68** | 19.90 / **0.65** | 23.80 / **0.65** |
| | CRS | **11.25 / 0.65** | 38.65 / 0.67 | **11.54** / **0.65** | **11.28 / 0.65** |

Table 15: Sampling-schedule dependence on LSUN church $256 \times 256$ in the latent-space diffusion model with stochastic samplers.

| sampler | NFE | sampling schedule | metrics FID ↓ | sFID ↓ | precision ↑ | recall ↑ |
|---|---|---|---|---|---|---|
| Stochastic DDIM | 100 | linear | 4.36 | 12.23 | 0.64 | 0.53 |
| | | cosine | 4.49 | 14.55 | **0.66** | 0.52 |
| | | EDM | 5.78 | 14.34 | 0.63 | 0.49 |
| | | CRS | **4.01** | **11.27** | 0.64 | **0.55** |
| | 50 | linear | 6.41 | 15.10 | 0.62 | 0.47 |
| | | cosine | 6.77 | 21.60 | **0.65** | 0.47 |
| | | EDM | 11.14 | 19.04 | 0.59 | 0.33 |
| | | CRS | **5.29** | **12.93** | 0.62 | **0.50** |
| | 30 | linear | 10.85 | 18.77 | 0.57 | 0.34 |
| | | cosine | 11.34 | 30.26 | **0.61** | **0.40** |
| | | EDM | 24.30 | 33.46 | 0.42 | 0.21 |
| | | CRS | **7.98** | **14.12** | 0.59 | 0.38 |
| SDE-DPM-Solver++(2M) | 100 | linear | 3.44 | 10.90 | 0.65 | 0.57 |
| | | cosine | 3.53 | 11.75 | **0.66** | 0.55 |
| | | EDM | 3.43 | 10.83 | 0.64 | 0.56 |
| | | CRS | **3.24** | **10.10** | 0.64 | **0.58** |
| | 50 | linear | 3.42 | 10.70 | 0.64 | 0.57 |
| | | cosine | 4.09 | 12.94 | **0.66** | 0.53 |
| | | EDM | 3.67 | 11.36 | 0.64 | 0.57 |
| | | CRS | **3.27** | **9.87** | 0.63 | **0.58** |
| | 30 | linear | **3.50** | 10.69 | 0.63 | **0.57** |
| | | cosine | 5.03 | 16.01 | **0.66** | 0.50 |
| | | EDM | 4.35 | 12.30 | 0.63 | 0.53 |
| | | CRS | 3.63 | **9.83** | 0.61 | **0.57** |

Table 16: Sampling-schedule dependence on LSUN church $256 \times 256$ in the latent-space diffusion model with deterministic samplers.

| sampler | NFE | sampling schedule | metrics FID ↓ | sFID ↓ | precision ↑ | recall ↑ |
|---------|-----|-------------------|------|--------|-------------|----------|
| DDIM | 100 | linear | 4.16 | 11.41 | **0.61** | 0.57 |
| | | cosine | 17.35 | 45.70 | 0.53 | 0.51 |
| | | EDM | 4.70 | 11.74 | 0.60 | 0.54 |
| | | CRS | **3.79** | **10.26** | 0.60 | **0.58** |
| | 50 | linear | 4.90 | 12.28 | **0.61** | 0.55 |
| | | cosine | 15.14 | 39.63 | 0.54 | 0.50 |
| | | EDM | 6.25 | 13.68 | 0.59 | 0.50 |
| | | CRS | **4.41** | **10.80** | 0.60 | **0.56** |
| | 30 | linear | 6.06 | 13.69 | **0.59** | 0.51 |
| | | cosine | 15.10 | 36.11 | 0.54 | 0.48 |
| | | EDM | 10.24 | 18.61 | 0.56 | 0.44 |
| | | CRS | **5.48** | **11.64** | 0.57 | **0.54** |
| PNDM | 100 | linear | 3.65 | 10.86 | **0.61** | 0.58 |
| | | cosine | 230.27 | 217.62 | 0.00 | 0.00 |
| | | EDM | 3.76 | 11.15 | **0.61** | 0.58 |
| | | CRS | **3.48** | **10.28** | **0.61** | **0.59** |
| | 50 | linear | 3.64 | 10.84 | **0.61** | 0.58 |
| | | cosine | 246.29 | 193.49 | 0.00 | 0.00 |
| | | EDM | 3.80 | 11.18 | **0.61** | 0.58 |
| | | CRS | **3.46** | **9.90** | **0.61** | **0.60** |
| | 30 | linear | **3.59** | 10.61 | **0.61** | 0.59 |
| | | cosine | 263.99 | 178.27 | 0.00 | 0.00 |
| | | EDM | 3.97 | 11.31 | 0.60 | 0.58 |
| | | CRS | 3.62 | **9.61** | 0.60 | **0.61** |
| DPM-Solver++(2M) | 100 | linear | 3.70 | 10.92 | **0.61** | 0.58 |
| | | cosine | 13.88 | 39.14 | 0.55 | 0.52 |
| | | EDM | 3.86 | 11.24 | **0.61** | 0.58 |
| | | CRS | **3.47** | **9.93** | 0.60 | **0.59** |
| | 50 | linear | 3.75 | 10.89 | **0.61** | 0.58 |
| | | cosine | 11.08 | 32.42 | 0.57 | 0.53 |
| | | EDM | 3.99 | 11.39 | 0.60 | 0.58 |
| | | CRS | **3.58** | **9.66** | 0.60 | **0.59** |
| | 30 | linear | **3.78** | 10.51 | **0.60** | **0.58** |
| | | cosine | 9.71 | 27.25 | 0.58 | 0.52 |
| | | EDM | 4.53 | 11.96 | **0.60** | 0.57 |
| | | CRS | 3.98 | **9.46** | 0.59 | **0.58** |

Table 17: Sampling-schedule dependence on LSUN bedroom $256 \times 256$ in the latent-space diffusion model with stochastic samplers.

| sampler | NFE | sampling schedule | metrics | | | |
|---|---|---|---|---|---|---|
| | | | FID ↓ | sFID ↓ | precision ↑ | recall ↑ |
| Stochastic DDIM | 100 | linear | 2.78 | 6.86 | 0.58 | 0.51 |
| | | cosine | 2.77 | 7.24 | 0.57 | 0.52 |
| | | EDM | 3.64 | 8.44 | 0.58 | 0.48 |
| | | CRS | **2.45** | **6.14** | **0.59** | **0.53** |
| | 50 | linear | 4.14 | 9.41 | 0.57 | 0.47 |
| | | cosine | 4.31 | 10.04 | 0.56 | 0.48 |
| | | EDM | 7.91 | 15.78 | 0.52 | 0.39 |
| | | CRS | **3.28** | **7.41** | **0.58** | **0.49** |
| | 30 | linear | 7.43 | 15.53 | 0.53 | 0.38 |
| | | cosine | 7.59 | 13.57 | 0.53 | 0.42 |
| | | EDM | 17.55 | 33.64 | 0.40 | 0.25 |
| | | CRS | **5.22** | **10.36** | **0.56** | **0.44** |
| SDE-DPM-Solver++(2M) | 100 | linear | 2.45 | 6.30 | **0.57** | 0.54 |
| | | cosine | 2.42 | 6.26 | 0.56 | 0.54 |
| | | EDM | 2.29 | 6.14 | **0.57** | 0.54 |
| | | CRS | **2.22** | **5.94** | **0.57** | **0.55** |
| | 50 | linear | 2.44 | 6.49 | **0.57** | 0.53 |
| | | cosine | 2.84 | 7.22 | 0.55 | 0.54 |
| | | EDM | 2.49 | 6.46 | **0.57** | 0.54 |
| | | CRS | **2.33** | **6.04** | **0.57** | **0.55** |
| | 30 | linear | **2.51** | 6.73 | **0.57** | **0.54** |
| | | cosine | 3.75 | 9.27 | 0.56 | 0.51 |
| | | EDM | 3.03 | 7.59 | 0.56 | 0.53 |
| | | CRS | 2.60 | **6.34** | **0.57** | **0.54** |

Table 18: Sampling-schedule dependence on LSUN bedroom $256 \times 256$ in the latent-space diffusion model with deterministic samplers.

| sampler | NFE | sampling schedule | metrics FID ↓ | sFID ↓ | precision ↑ | recall ↑ |
|---|---|---|---|---|---|---|
| DDIM | 100 | linear | 2.77 | 6.34 | 0.54 | 0.55 |
| | | cosine | 9.59 | 15.74 | 0.45 | **0.57** |
| | | EDM | 3.17 | 6.74 | 0.53 | 0.55 |
| | | CRS | **2.57** | **6.01** | **0.55** | 0.55 |
| | 50 | linear | 3.20 | 6.83 | 0.53 | 0.54 |
| | | cosine | 8.38 | 13.88 | 0.46 | **0.56** |
| | | EDM | 4.20 | 7.75 | 0.52 | 0.51 |
| | | CRS | **2.82** | **6.24** | **0.54** | 0.54 |
| | 30 | linear | 3.89 | 7.73 | 0.52 | 0.51 |
| | | cosine | 8.87 | 13.91 | 0.45 | 0.53 |
| | | EDM | 6.95 | 10.24 | 0.47 | 0.47 |
| | | CRS | **3.34** | **6.76** | **0.54** | **0.52** |
| PNDM | 100 | linear | 2.59 | 6.12 | **0.54** | 0.56 |
| | | cosine | 248.80 | 213.86 | 0.31 | 0.00 |
| | | EDM | 2.64 | 6.20 | 0.53 | **0.57** |
| | | CRS | **2.54** | **5.94** | **0.54** | 0.56 |
| | 50 | linear | 2.59 | 6.16 | **0.54** | **0.57** |
| | | cosine | 245.17 | 157.84 | 0.15 | 0.00 |
| | | EDM | 2.70 | 6.26 | 0.53 | **0.57** |
| | | CRS | **2.54** | **5.77** | **0.54** | **0.57** |
| | 30 | linear | **2.65** | 6.18 | 0.53 | **0.57** |
| | | cosine | 254.17 | 128.18 | 0.04 | 0.00 |
| | | EDM | 3.00 | 6.58 | 0.52 | 0.56 |
| | | CRS | 2.75 | **5.73** | 0.53 | **0.57** |
| DPM-Solver++(2M) | 100 | linear | 2.63 | 6.23 | **0.54** | 0.56 |
| | | cosine | 7.73 | 13.34 | 0.47 | **0.57** |
| | | EDM | 2.71 | 6.29 | 0.53 | **0.57** |
| | | CRS | **2.51** | **5.92** | **0.54** | **0.57** |
| | 50 | linear | 2.68 | 6.29 | 0.54 | **0.57** |
| | | cosine | 6.33 | 11.16 | **0.58** | **0.57** |
| | | EDM | 2.93 | 6.55 | 0.53 | **0.57** |
| | | CRS | **2.51** | **5.72** | **0.54** | 0.56 |
| | 30 | linear | 2.81 | 6.34 | **0.53** | **0.56** |
| | | cosine | 5.97 | 10.25 | 0.48 | **0.56** |
| | | EDM | 3.34 | 6.91 | **0.53** | 0.55 |
| | | CRS | **2.70** | **5.80** | **0.53** | **0.56** |

Table 19: Sampling-schedule dependence on ImageNet $256 \times 256$ in the latent-space diffusion model with stochastic samplers.

| | | sampling | metrics | | | |
|---|---|---|---|---|---|---|
| sampler | NFE | schedule | FID ↓ | sFID ↓ | precision ↑ | recall ↑ |
| Stochastic DDIM | 100 | linear | 10.98 | 5.94 | 0.62 | **0.62** |
| | | cosine | 12.13 | 13.11 | 0.60 | **0.62** |
| | | EDM | 13.33 | 8.04 | 0.59 | 0.61 |
| | | CRS | **10.15** | **5.53** | **0.63** | **0.62** |
| | 50 | linear | 15.10 | 9.96 | 0.57 | 0.60 |
| | | cosine | 15.41 | 17.70 | 0.56 | **0.61** |
| | | EDM | 25.28 | 20.79 | 0.47 | 0.57 |
| | | CRS | **12.89** | **7.48** | **0.60** | 0.60 |
| | 30 | linear | 25.50 | 21.41 | 0.48 | 0.56 |
| | | cosine | 23.22 | 31.04 | 0.48 | **0.58** |
| | | EDM | 48.93 | 52.41 | 0.32 | 0.51 |
| | | CRS | **20.48** | **14.89** | **0.53** | 0.57 |
| SDE-DPM-Solver++(2M) | 100 | linear | 9.21 | 5.41 | 0.63 | **0.65** |
| | | cosine | 9.73 | 9.08 | 0.62 | 0.63 |
| | | EDM | **8.93** | **5.30** | **0.64** | 0.64 |
| | | CRS | 9.02 | 5.41 | **0.64** | 0.64 |
| | 50 | linear | **9.22** | 5.57 | 0.63 | **0.64** |
| | | cosine | 10.47 | 10.11 | 0.61 | 0.63 |
| | | EDM | 9.25 | 5.44 | 0.63 | **0.64** |
| | | CRS | 9.34 | **5.33** | **0.64** | 0.63 |
| | 30 | linear | **10.22** | 5.81 | 0.62 | **0.64** |
| | | cosine | 11.77 | 9.37 | 0.61 | 0.62 |
| | | EDM | 11.17 | 6.56 | 0.61 | 0.63 |
| | | CRS | 10.25 | **5.49** | **0.63** | 0.63 |

Table 20: Sampling-schedule dependence on ImageNet $256 \times 256$ in the latent-space diffusion model with deterministic samplers.

| sampler | NFE | sampling schedule | metrics FID ↓ | sFID ↓ | precision ↑ | recall ↑ |
|---|---|---|---|---|---|---|
| DDIM | 100 | linear | 11.91 | **5.33** | **0.59** | 0.65 |
| | | cosine | 44.94 | 66.68 | 0.32 | **0.67** |
| | | EDM | 12.73 | 5.44 | **0.59** | 0.65 |
| | | CRS | **11.72** | 6.18 | **0.59** | 0.65 |
| | 50 | linear | 13.08 | **5.71** | **0.58** | 0.64 |
| | | cosine | 35.56 | 48.27 | 0.35 | **0.66** |
| | | EDM | 15.43 | 6.54 | 0.56 | 0.64 |
| | | CRS | **13.03** | 7.00 | **0.58** | 0.64 |
| | 30 | linear | 15.52 | **6.95** | **0.56** | 0.63 |
| | | cosine | 33.80 | 39.15 | 0.36 | **0.65** |
| | | EDM | 20.92 | 9.78 | 0.50 | 0.62 |
| | | CRS | **15.17** | 8.65 | **0.56** | 0.62 |
| PNDM | 100 | linear | 11.16 | **5.31** | **0.60** | 0.65 |
| | | cosine | 176.01 | 132.61 | 0.09 | 0.00 |
| | | EDM | 11.43 | **5.31** | **0.60** | **0.66** |
| | | CRS | **11.00** | 5.58 | **0.60** | 0.65 |
| | 50 | linear | 11.24 | **5.26** | **0.60** | **0.66** |
| | | cosine | 170.24 | 107.05 | 0.11 | 0.00 |
| | | EDM | 11.63 | 5.27 | **0.60** | **0.66** |
| | | CRS | **11.10** | 5.62 | **0.60** | 0.65 |
| | 30 | linear | 11.42 | **5.31** | 0.59 | **0.66** |
| | | cosine | 172.05 | 79.53 | 0.10 | 0.00 |
| | | EDM | 11.97 | 5.39 | 0.59 | **0.66** |
| | | CRS | **11.38** | 5.66 | **0.60** | 0.65 |
| DPM-Solver++(2M) | 100 | linear | **11.25** | **5.33** | **0.60** | 0.65 |
| | | cosine | 38.65 | 58.04 | 0.35 | **0.67** |
| | | EDM | 11.54 | 5.34 | **0.60** | 0.65 |
| | | CRS | 11.28 | 5.55 | **0.60** | 0.65 |
| | 50 | linear | **11.59** | **5.33** | 0.59 | 0.65 |
| | | cosine | 29.48 | 38.19 | 0.40 | **0.66** |
| | | EDM | 12.00 | 5.36 | 0.59 | **0.66** |
| | | CRS | 11.62 | 5.53 | **0.60** | 0.65 |
| | 30 | linear | **12.09** | **5.40** | 0.59 | 0.65 |
| | | cosine | 25.48 | 26.64 | 0.42 | **0.66** |
| | | EDM | 12.77 | 5.58 | 0.58 | **0.66** |
| | | CRS | 12.22 | 5.62 | **0.60** | 0.64 |

Table 21: Sampling-schedule dependence on LSUN horse $256 \times 256$ in the pixel-space diffusion model.

| sampler | NFE | sampling schedule | metrics FID ↓ | sFID ↓ | precision ↑ | recall ↑ |
|---|---|---|---|---|---|---|
| SDE-DPM-Solver++(2M) | 250 | linear | 2.86 | 6.60 | 0.66 | **0.56** |
| | | cosine | 3.09 | 7.13 | **0.68** | **0.56** |
| | | EDM | 2.34 | 6.47 | **0.68** | **0.56** |
| | | CRS | **2.30** | **6.34** | **0.68** | **0.56** |
| PNDM | 250 | linear | 3.90 | 6.23 | 0.57 | **0.61** |
| | | cosine | 3.00 | 5.46 | 0.61 | 0.60 |
| | | EDM | 2.85 | **5.66** | **0.63** | 0.58 |
| | | CRS | **2.67** | 5.69 | **0.63** | 0.59 |
| DPM-Solver++(2M) | 250 | linear | 3.06 | 5.96 | 0.60 | **0.60** |
| | | cosine | 2.72 | 5.58 | 0.62 | **0.60** |
| | | EDM | 2.83 | 5.52 | 0.63 | 0.58 |
| | | CRS | **2.68** | **5.36** | **0.64** | 0.59 |

Table 22: Sampling-schedule dependence on LSUN cat $256 \times 256$ in the pixel-space diffusion model.

| sampler | NFE | sampling schedule | metrics | | | |
| --- | --- | --- | --- | --- | --- | --- |
| | | | FID ↓ | sFID ↓ | precision ↑ | recall ↑ |
| SDE-DPM-Solver++(2M) | 250 | linear | 6.35 | 7.20 | 0.59 | 0.49 |
| | | cosine | 7.92 | 7.73 | 0.56 | 0.52 |
| | | EDM | 5.26 | 7.09 | **0.60** | **0.53** |
| | | CRS | **5.25** | **6.48** | **0.60** | **0.53** |
| PNDM | 250 | linear | 7.67 | 7.07 | 0.47 | **0.57** |
| | | cosine | 6.43 | 6.02 | 0.52 | 0.56 |
| | | EDM | 5.92 | 6.08 | 0.54 | 0.56 |
| | | CRS | **5.51** | **5.71** | **0.55** | 0.56 |
| DPM-Solver++(2M) | 250 | linear | 6.22 | 6.33 | 0.50 | **0.57** |
| | | cosine | 5.83 | 5.81 | 0.53 | **0.57** |
| | | EDM | 5.89 | 5.99 | 0.54 | 0.56 |
| | | CRS | **5.58** | **5.64** | **0.55** | 0.56 |

training: linear / sampling: linear / FID=3.51

training: linear / sampling: CRS / FID=3.22

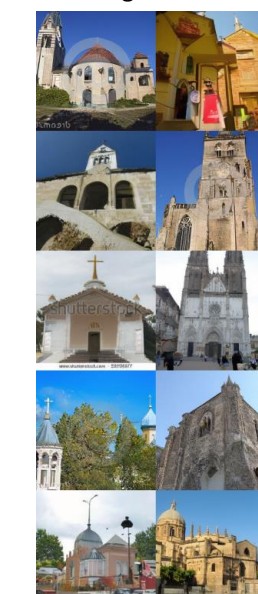

training: CRS / sampling: linear / FID=3.44

training: CRS / sampling: CRS / FID=3.24

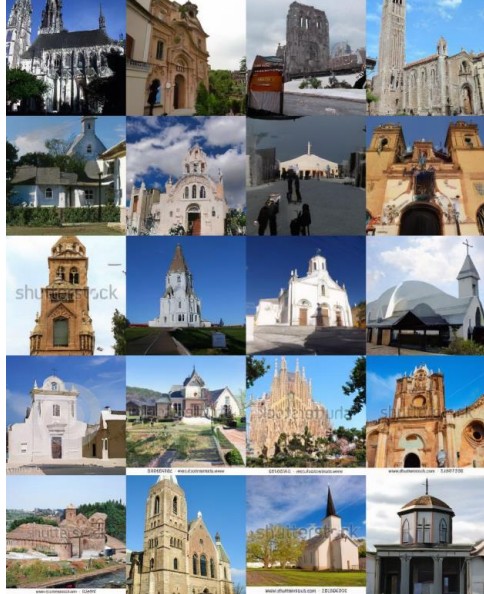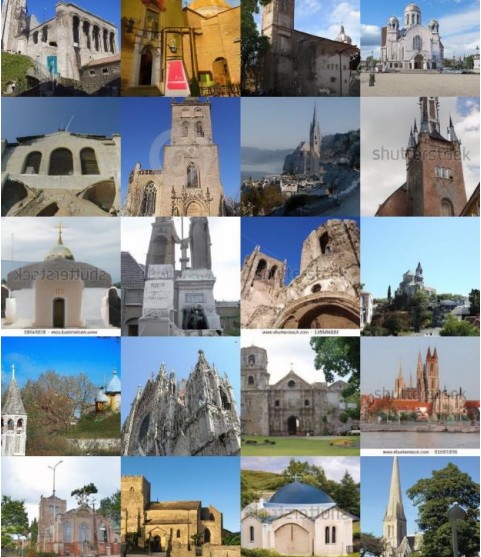

Figure 4: Samples of LSUN church 256 × 256 generated using our latent-space diffusion model with SDE-DPM-Solver++(2M) at NFE=100.

training: CRS / sampling: linear / FID=2.44

training: CRS / sampling: cosine / FID=2.84

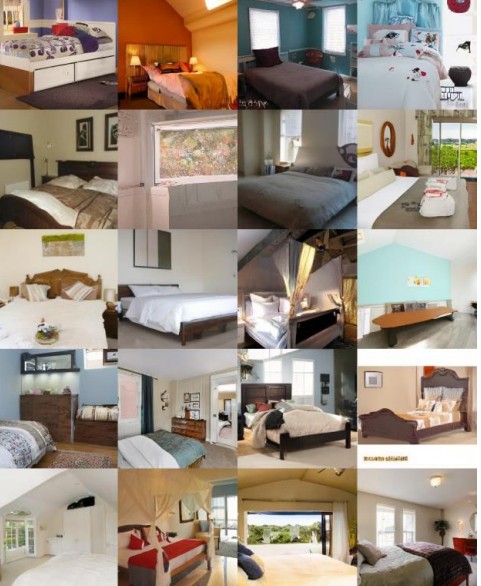

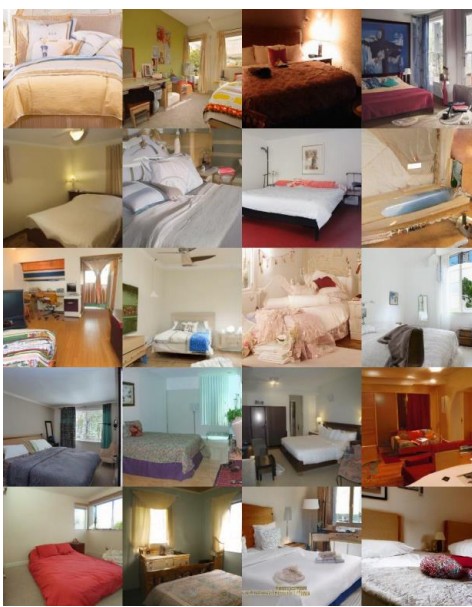

training: CRS / sampling: EDM / FID=2.49

training: CRS / sampling: CRS / FID=2.33

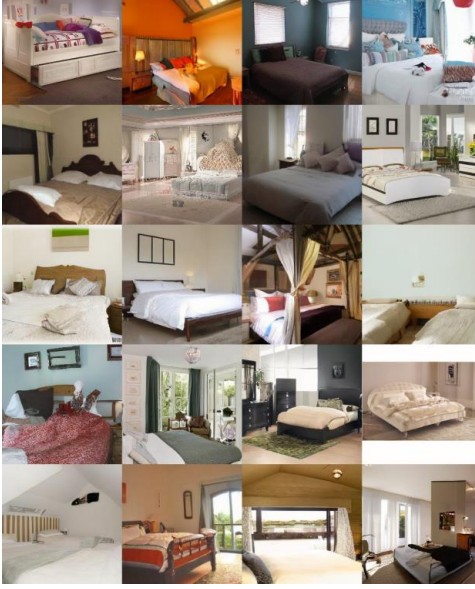

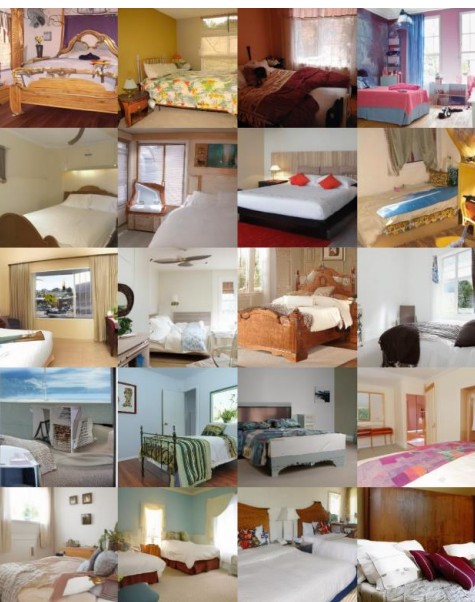

Figure 5: Samples of LSUN bedroom $256 \times 256$ generated using our latent-space diffusion model with SDE-DPM-Solver++(2M) at NFE=50.

training: linear / sampling: linear / FID=10.67

training: linear / sampling: CRS / FID=10.84

training: CRS / sampling: linear / FID=9.21

training: CRS / sampling: CRS / FID=9.02

Figure 6: Samples of ImageNet $256 \times 256$ generated using our latent-space diffusion model with SDE-DPM-Solver++(2M) at NFE=100. Five classes are randomly selected, and images in the same row belong to the same class.

training: linear / sampling: linear / FID=10.67    training: linear / sampling: CRS / FID=10.84

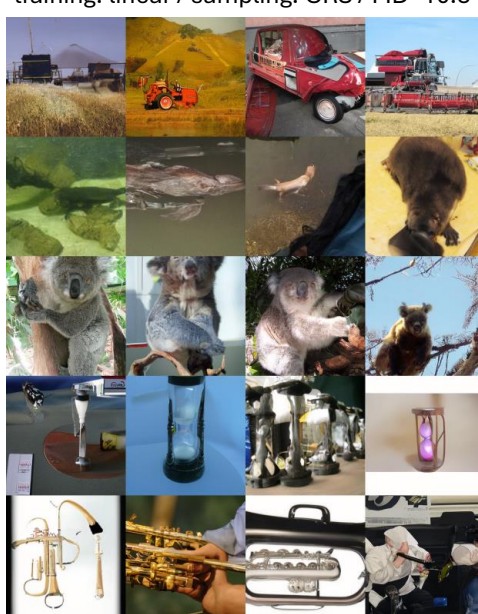

training: CRS / sampling: linear / FID=9.21    training: CRS / sampling: CRS / FID=9.02

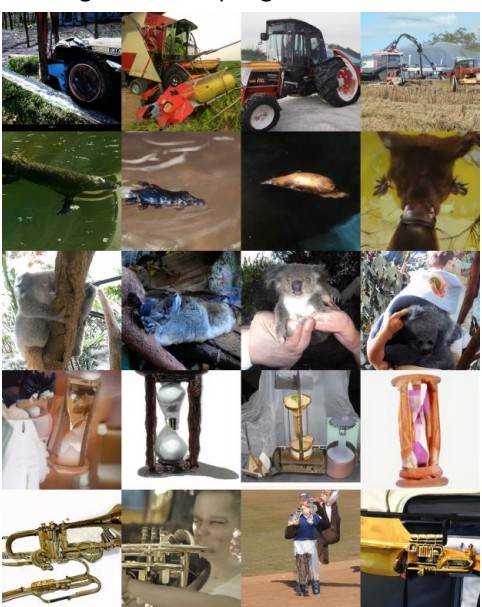
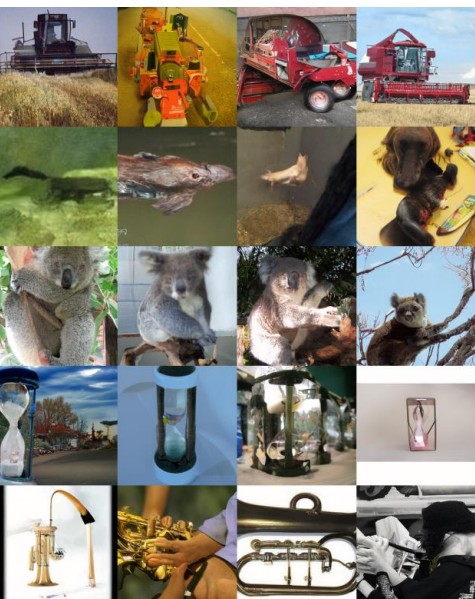

Figure 7: Samples of ImageNet 256 × 256 generated using our latent-space diffusion model with SDE-DPM-Solver++(2M) at NFE=100. Five classes are randomly selected, and images in the same row belong to the same class.

training: linear / sampling: linear / FID=2.90   training: linear / sampling: cosine / FID=3.09

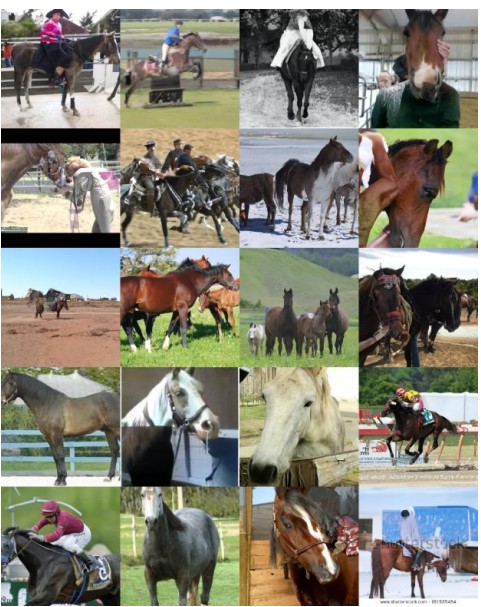

training: linear / sampling: EDM / FID=2.34   training: linear / sampling: CRS / FID=2.30

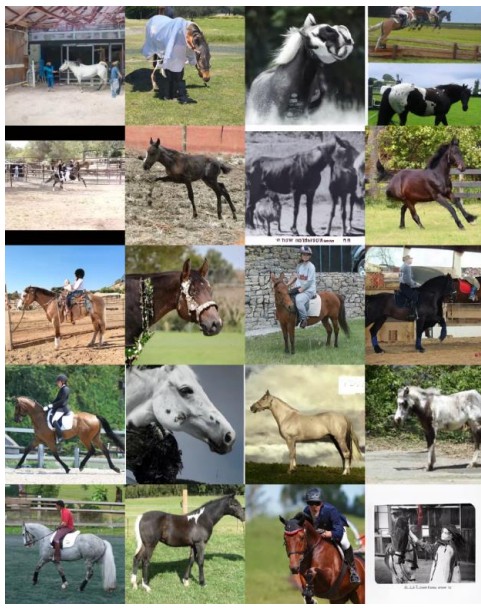
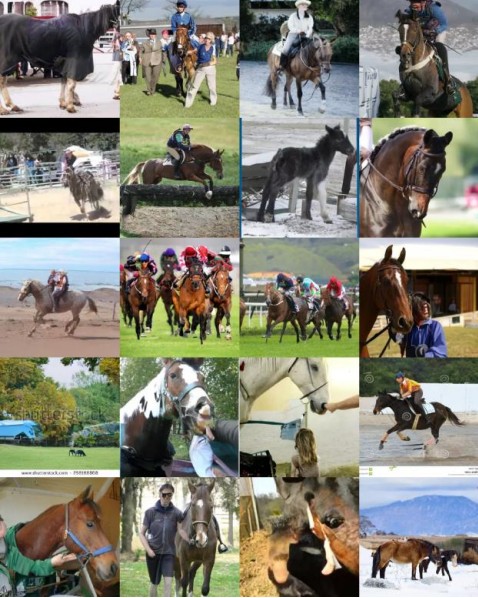

Figure 8: Samples of LSUN horse $256 \times 256$ generated using our pixel-space diffusion model with SDE-DPM-Solver++(2M) at NFE=250.

training: linear / sampling: linear / FID=6.35     training: linear / sampling: cosine / FID=7.92

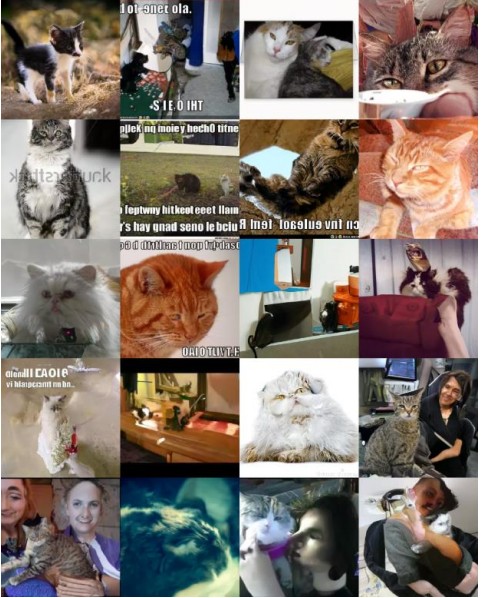

training: linear / sampling: EDM / FID=5.26     training: linear / sampling: CRS / FID=5.25

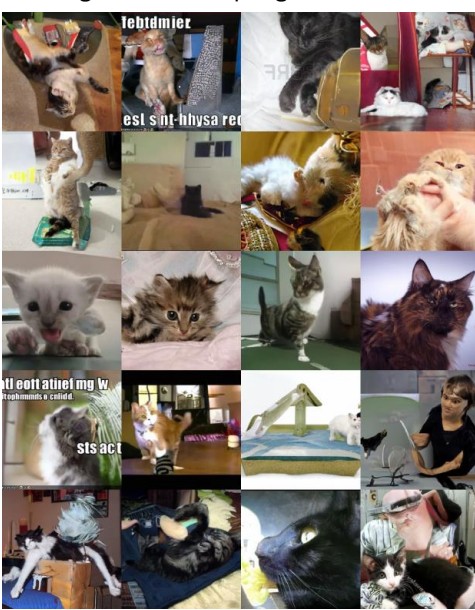
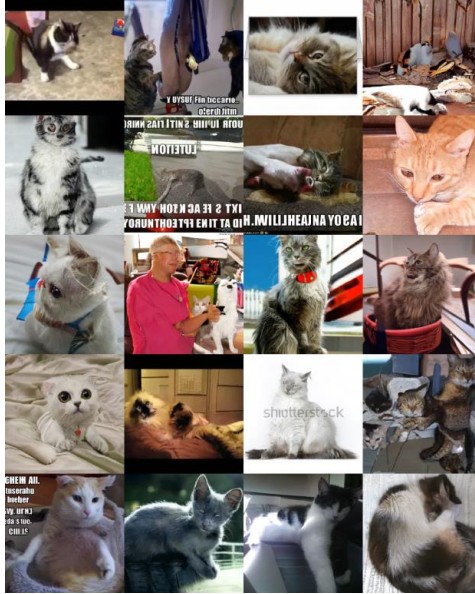

Figure 9: Samples of LSUN cat $256 \times 256$ generated using our pixel-space diffusion model with SDE-DPM-Solver++(2M) at NFE=250.

