# OpenReview forum: "Constant Rate Schedule: Constant-Rate Distributional Change for Efficient Training and Sampling in Diffusion Models"
_ICLR.cc/2025/Conference — ICLR 2025 Conference Withdrawn Submission_

### Official Review · Reviewer_Bp1L · 2024-10-22

**Soundness:** 1
**Presentation:** 2
**Contribution:** 1
**Rating:** 3
**Confidence:** 4

**Summary:**

This paper presents the Constant Rate Schedule (CRS) to enhance the sampling speed of diffusion models. CRS aims to ensure a constant rate of change in the probability distribution of diffused data. It determines the noise schedule by minimizing the maximum distance between consecutive probability distributions in the forward process. The authors perform experiments on multiple datasets and models show that compared to some baseline schedules, CRS enhances performance in latent-space diffusion models and when employed as a sampling schedule in pixel-space models.

**Strengths:**

The authors propose a simple algorithm for automatically deriving a noise schedule for diffusion models and present some relevant experimental results.

**Weaknesses:**

1. The algorithmic design of this work exhibits several drawbacks. It is overly simplistic, relying solely on the FID as the distance metric for computing $\tilde{D}(\alpha,\alpha')$ without exploring other potentially more suitable options. Additionally, the authors fail to present a clear and comprehensive pipeline for their algorithm, leaving readers with an incomplete understanding of its implementation details.

2. The experimental results are subpar. In the current landscape of diffusion models, numerous advanced sampling methods have emerged. For instance, UniPC (https://arxiv.org/abs/2302.04867) and DPM-Solver-v3 (https://arxiv.org/abs/2310.13268) can achieve high-quality generation in as few as 10 sampling steps, and even DMD (https://arxiv.org/abs/2311.18828) and DMD2 (https://arxiv.org/abs/2405.14867) can generate satisfactory results in a single step. However, this work predominantly reports results for cases where the number of function evaluations (NFE) is larger than or equal 50. This focus is of limited relevance when the aim is to enhance the sampling speed of diffusion models. Moreover, in the case of using 30 NFEs for ImageNet-256 (as shown in Tables 19 and 20), the obtained FID is considerably higher than that achieved by DPM-Solver-v3 with only 10 NFEs (as evident from their Table 6).

3. There are issues with the mathematical derivations. In Line 179, the authors state that the derivation of the reverse SDE and ODE of the variance-preserving process can be found in Appendix C. However, Appendix C merely replicates content from other papers (ScoreSDE and EDM) without any additional explanation or modification. It is unclear why the authors removed the dependence on $t$ in Eqs. (8) and (9) (and Eq. (10) is a null equation).

In light of the aforementioned weaknesses, I have the impression that this submission clearly falls short of the standards expected for ICLR.

**Questions:**

No.

---

### Official Review · Reviewer_sWbG · 2024-10-29

**Soundness:** 2
**Presentation:** 2
**Contribution:** 2
**Rating:** 3
**Confidence:** 4

**Summary:**

In this paper, the authors present a new method to obtain a noise schedule that depends on a given dataset in the context of diffusion models. The procedure works as follows. The authors compute an approximation of the derivative of some distance (FID) between noisy samples with respect to some schedule parameter. Once this is done the authors design a noise schedule such that this derivative is constant. They evaluate their methodology on several benchmark datasets and compare it with other existing methodologies to optimize the schedule such as Align Your Steps [1]. They report better FID numbers than baselines on several datasets.

[1] Sabour et al. – Align Your Steps: Optimizing Sampling Schedules in Diffusion Models

**Strengths:**

* To the best of my knowledge the idea is original. The paper is fairly well written and the main concepts are presented in a concise manner.

* I think such an approach might interest the community. Optimizing the schedule in diffusion models is often overlooked by existing approaches. There is untapped potential there and I think this work might lead to further research in that direction.

**Weaknesses:**

* One of the main limitations of the paper is the lack of details on the computation of the distance and its derivative which is absolutely central to the paper. I had a look at Appendix B but could not find satisfactory answers to many of my questions. For example, what is the overhead of computing all those distances? One has to compute the InceptionV3 features at many noise levels which might be quite expensive as there is no easy way to compute the InceptionV3 features. I am also wondering what is the significance of computing FID scores between noisy images. Sure FID score captures some information about the distributions and their misalignment but InceptionV3 was trained on natural images so I am questioning what really is measured when the noise level is positive. To summarize I think the authors need to provide more details around their main contributions, how to compute it and how to motivate it.

* I am a bit worried about the dependency with the hyperparameter $\xi$. This dependency is introduced and then never discussed but it changes values (see Table 12). This defeats the purpose of the paper as one of the goals is to derive adaptive schedules and reduce the number of hyperparameters but a new hyperparameter is introduced and not discussed. Changing this hyperparameter from the value of $1$ also changes the motivation of the work since we no longer have a constant rate schedule.

**Questions:**

* “We find that the efficient noise schedule depends on both the target dataset and the type of diffusion model. In particular, the efficient noise schedule in latent-space diffusion models is very different from conventionally used ones.” I don’t think the schedule identified is very different from the commonly used ones if you include Stochastic Interpolant and Flow Matching types of approaches. In that case the schedule is given by $\alpha_t = 1-t$ and $\sigma_t = t$ (which doesn’t fit in the framework studied by the authors since $\sigma_t^2 \neq 1- \alpha_t^2$. However, I would suggest taming the claims here since very similar schedules to the one identified have been investigated.

* There are some repeated entries in the bibliography (l.588 for instance)

* Looking at Table 10 it seems that the weighting function used in the current work for the training of EDM is different than the one used in EDM. Why is that? For fairness it would be good to train EDM with the same schedule.

* No results are reported on CIFAR10. Given that EDM reports strong results on CIFAR10 it would be good to see what is the influence of CRS in that setting compared to the EDM baseline.

* I think there is a typo in Equation (17) it should be $-\xi$ and not $\xi$.

---

### Official Review · Reviewer_Y3P9 · 2024-11-01

**Soundness:** 3
**Presentation:** 1
**Contribution:** 2
**Rating:** 3
**Confidence:** 3

**Summary:**

This paper introduces a noise scheduling method called the Constant Rate Schedule (CRS) for diffusion models, designed to enhance training and sampling efficiency. The authors propose a principled approach that adaptively selects noise schedules based on the dataset and model architecture, eliminating the need for a predefined functional form. Inspired by the adiabatic theorem in quantum annealing, CRS ensures a constant rate of distributional change at each step of the diffusion process, according to a user-specified metric (or divergence) in the space of probability distributions. Using Fr\'echet Inception Distance (FID) as the metric, the paper validates CRS through extensive numerical experiments, demonstrating that the proposed CRS can improve the performance of diffusion models.

**Strengths:**

This work contributes meaningfully to improving diffusion model efficiency by introducing an adaptive noise scheduling method that systematically adjusts the diffusion process.  While the idea of adaptive noise scheduling is not surprising, the authors make a novel contribution by explicitly deriving a scheduling procedure by leveraging constant-rate distributional change and the proposed CRS scheme is thoroughly validated with extensive experimentation.  The results indicate CRS’s effectiveness across different datasets, samplers, and scheduling settings, highlighting its efficacy and broad applicability.  The generality of CRS suggests it could have broader applications beyond image generation, indicating its significance and further potential impact.

**Weaknesses:**

Despite the paper's sound methodological contributions and promising experimental results, its presentation can benefit from substantial refinement for clarity and reproducibility.  Specific issues and recommendations are as follows.

**1. Methodology presentation**

Section 4 would benefit from a clearer, more structured presentation of the CRS procedure, potentially in a pseudocode format. The lack of an explicit procedural outline with well-defined inputs, outputs, and step-by-step operations makes the method challenging for the readers to comprehend and reproduce.  Clarifying this detail in the main text, rather than relying on Appendix B (indeed, I found some aspects remaining not very clear even after referring to the appendix), would provide greater transparency.  The presentation could also improve by reorganizing Section 4 to enhance logical flow and readability.  For instance, the authors can augment the procedure described in Lines 265 -- 269 into a concise pseudocode format and include it in Section 4.2, and maybe the authors may want to merge Sections 4.3 & 4.4 into "Additional considerations" etc.

**2. Presentation of Experimental Results**

In Section 5, the paper could reduce redundancy and make room for more critical content by moving some tables to the Appendix and bringing essential content, such as Algorithm 1 (which can be potentially converted into the pseudocode to include in Section 4), into the main text. Consolidating Tables 10 and 11 in the main text, while moving repetitive tables to the Appendix, would help readers focus on the primary results. Additionally, defining key metrics like FID, sFID, precision, and recall would support readers unfamiliar with diffusion models.
* Regarding latent-space diffusion models, a brief summary or introduction to these models would be beneficial for accessibility.
* In Section 5.2, a conjecture or discussion on why CRS showed limited improvement in pixel-space models as a training schedule could provide valuable insights.
* The rationale for selecting the hyperparameter $\xi$ may need to be explained in more detail.  For instance, why did the authors choose \xi = 1.2 in Line 465, while the informal argument in Section 4.2 suggest \xi = 1?  Also, the authors may want to give a better, more detailed justification in Line 251, why they introduce a hyperparameter \xi (not necessarily 1) for a more flexibile modeling/training.

**3. Clarity & Logical Flow**
* *Summary of contributions*: The stated contributions in Lines 90–103 could be revised for clarity and cohesion. In particular, Item 1’s reference to the “type” of diffusion model could be clarified (e.g., by specifying architecture or training objective). Additionally, Item 2 appears to be a detail that complements Item 1 rather than an independent contribution, which could enhance the readability of this section.
* *Discussion and Conclusion*:  The Discussion and Conclusion sections could be consolidated to provide a cohesive summary of findings, limitations, and future directions. Alternatively, if the authors intended Section 6 for experimental analysis, integrating it into Section 5 would streamline the paper’s structure.

**4. Minor suggestions**
* Acronyms, such as CRS (Line 47) and AYS (Line 135), should be defined with their full names upon first usage. Additionally, vague notation (e.g., $\alpha$ in Lines 53-54, which is not defined or explained) may require further definition and context. Also, readers may find the distinction among $\alpha$ thresholds (e.g., 0.6 and 0.97) ambiguous and unclear whether they are universal or example-specific -- further, the distinction between the "second" and the "third" is not very clear to me.
* In Lines 78–79, the text would benefit from mentioning earlier that the CRS depends on the choice of a probability metric or divergence and is universally applicable to orient the reader.
* There are minor clarity issues, such as ambiguous conjunctions (e.g., "However" in Line 255 and "though" in Line 485). These can create logical dissonance, and revising these phrases for clarity would improve readability.
* It would be helpful to add definitions for FID, sFID, precision, and recall metrics, perhaps in the Appendix, to assist readers unfamiliar with these evaluation tools.

**Questions:**

Here I list a few concrete questions/suggestions, extracted from my comments in "Weaknesses."

**1. Precise Description of Methodology for Reproducibility:**
Could you provide a more detailed, step-by-step description of the CRS algorithm in Section 4? A pseudocode with inputs and outputs explicitly defined would be beneficial.

**2. Hyperparameter $\xi$:**
Could you provide additional context for choosing $\xi$ values, especially given the variation between $\xi=1.0$ in Section 4.2 and $\xi=1.2$ in some experiments (e.g., in Section 5.2)?  Since the informal argument in Section 4.2 (esp. Eq. (15)) suggests $\xi=1.0$, it would be beneficial to discuss why the authors wanted to introduce a hyperparameter $\xi$ for more flexibility, how they chose the specific values in the experiments, and if they have any suggestions for practitioners about how to choose $\xi$.

**3. Insights on Limited Benefits in the Pixel-space Diffusion Models:**
Do the authors have any insights or hypotheses for why CRS showed limited effectiveness in training pixel-space diffusion models, as remarked in Section 5.2 (Lines 430 -- 431)?

---

### Official Review · Reviewer_c28o · 2024-11-01

**Soundness:** 2
**Presentation:** 2
**Contribution:** 2
**Rating:** 5
**Confidence:** 3

**Summary:**

- In the design of the diffusion models, we usually need to fix the noise scheduling by hand, and the choice of the scheduling is directly related to the performance of the resultant trained models.
- In this paper, the authors propose an almost automatic noise scheduling determination mechanism by making the distance between model distributions on diffusion time t and t+dt as constant.
    - They employ Fréchet Inception Distance in the paper.
- They perform some numerical experiments and show the effects of the proposals with:
    - LSUN church 256^2, LSUN bedroom 256^2, ImageNet 256^2 based on latent-space diffusion models,
    - LSUN horse 256^2, LSUN cat 256^2 based on pixel-space diffusion models.

**Strengths:**

- +originality, significance
    - The idea is simple and the results look good, at least the proposed methods provide better results compared to the baselines.

**Weaknesses:**

- -quality, clarity
    - But I have some concerns related to quality and clarity as summarized below.

**Questions:**

Comments
1. In line 123, the wording "score-based matching" is not common.

Questions
1. To compute the scheduling function, I guess we need to compute the derivative, eq (14), with D = FID, but I have no idea how to compute derivative of FID which is basically defined by point cloud. How could we compute it?
2. Related to the above question, the authors show the use of the proposed automatic scheduling during the training also. Does it mean we need to compute FID whenever we compute the loss function during the training? If so, it sounds a little time consuming. Are there any computational difficulties on it in practice?
3. How should we determine $\xi$ in eq (16)?
4. The scores in all tables show quite, maybe too strong results of the proposed method. I know there are some intuitive explanations on the proposed method in page 2, however I would like to know why it does improve the performance. Why does the "constant rate schedule" improve the scores so impressively? Is there any mathematical explanation on it?

---

### Note · Authors · 2024-11-13

I have read and agree with the venue's withdrawal policy on behalf of myself and my co-authors.